# Plant genetic diversity affects multiple trophic levels and trophic interactions

Nian-Feng Wan [1,8] ✉, Liwan Fu [2,3,8], Matteo Dainese [4], Yue-Qing Hu [3], Lars Pødenphant Kiær [5], Forest Isbell [6] & Christoph Scherber [7]

Intraspecific genetic diversity is an important component of biodiversity. A substantial body of evidence has demonstrated positive effects of plant genetic diversity on plant performance. However, it has remained unclear whether plant genetic diversity generally increases plant performance by reducing the pressure of plant antagonists across trophic levels for different plant life forms, ecosystems and climatic zones. Here, we analyse 4702 effect sizes reported in 413 studies that consider effects of plant genetic diversity on trophic groups and their interactions. We found that that increasing plant genetic diversity decreased the performance of plant antagonists including invertebrate herbivores, weeds, plant-feeding nematodes and plant diseases, while increasing the performance of plants and natural enemies of herbivores. Structural equation modelling indicated that plant genetic diversity increased plant performance partly by reducing plant antagonist pressure. These results reveal that plant genetic diversity often influences multiple trophic levels in ways that enhance natural pest control in managed ecosystems and consumer control of plants in natural ecosystems for sustainable plant production.

Plant genetic diversity has been shown to provide multiple ecosystem services (e.g. biological control[1]) in terrestrial[1–5] and marine[6] ecosystems. In natural ecosystems, plant genetic diversity is a key component of species and ecosystem health and can be manipulated by modifying the population genetic richness of plants within or around managed lands[1,2,7] or aquatic habitats[8]. In particular, increasing plant genetic diversity can increase plant productivity[9,10] and crop yield[1], suppress plant antagonists such as insect herbivores[11], weeds (i.e. harmful plants in human-controlled settings such as farm fields)[12], plant diseases[2,13] or plant-feeding nematodes[14], and promote natural enemies of pests[15,16]. One of the most important reasons that plant genetic diversity can affect trophic groups is due to intraspecific variation of plant chemistry within a species[17]. Single genotypes in monocultures may

be associated with reduced crop yields[18], increased pressure from herbivores[19] and plant diseases (i.e., pathogenic bacteria, fungi, and viruses)[20], and support fewer natural enemies of invertebrate herbivores[21] compared to more genetically diverse cropping systems. Together, these findings indicate a possible influence of plant genetic diversity that extends across multiple interacting trophic levels. However, a global synthesis on the effects of plant genetic diversity across trophic levels covering different plant life forms, ecosystems or climatic zones is still lacking, leaving it unclear whether results are consistent or context dependent.

Trophic interactions among plants, plant antagonists (e.g., invertebrate herbivores and plant pathogens) and natural enemies of herbivores (e.g., predators and parasitoids) are universal in nature[22,23] and

[1]Shanghai Key Laboratory of Chemical Biology, School of Pharmacy, East China University of Science and Technology, Shanghai 200237, China. [2]Center for Non-communicable Disease Management, Beijing Children's Hospital, Capital Medical University, National Center for Children's Health, Beijing 100045, China. [3]State Key Laboratory of Genetic Engineering, Institute of Biostatistics, School of Life Sciences, Fudan University, Shanghai 200438, China. [4]Department of Biotechnology, University of Verona, Verona, Italy. [5]Department of Plant and Environmental Sciences, University of Copenhagen, DK-1871 Frederiksberg C, Denmark. [6]Department of Ecology, Evolution and Behavior, University of Minnesota, Saint Paul, MN 55108, USA. [7]Centre for Biodiversity Monitoring and Conservation Science, Leibniz Institute for the Analysis of Biodiversity Change, Museum Koenig, Adenauerallee 127, 53113 Bonn, Germany. [8]These authors contributed equally: Nian-Feng Wan, Liwan Fu. ✉e-mail: nfwan@ecust.edu.cn

their role in structuring ecological communities has been extensively recognised (e.g. Underwood et al., 2017)[24]. For example, theory on trophic interactions predicts that herbivore natural enemies may indirectly increase plant performance by feeding on herbivores[25,26], and many studies have shown that plant genetic diversity strengthens bottom-up effects on higher trophic groups[16,27]. This is realised through increases in the abundance or diversity of predators and parasitoids of insect herbivores[9,28], decreases in the damage or abundance of herbivorous and nematode pests[11,14], decreased damage by plant pathogens[29], and mediation through plant chemistry as both primary and secondary plant metabolites are key mechanisms driving direct and indirect effects on upper trophic levels[17]. This generally results in increased plant growth and reproduction[9] at higher genetic diversity. However, several other studies have found results different from those described above[11,30–32].

A generalised literature synthesis is therefore needed to illustrate the gaps and trends in the research based on the existing literature, and to determine whether effect sizes are generalisable or context dependent[33,34]. Previous meta-analyses have concluded that plant genetic diversity promotes crop yields[18], increases abundance and species richness of predators[21], but has no significant effects on herbivore abundance or herbivory damage[21]. However, these syntheses involved only selected trophic groups (crops[18], or arthropod herbivores and their natural enemies[21]) and selected study systems (crop species) and did not consider other harmful plant antagonists such as weeds, plant diseases or plant-feeding nematodes. Moreover, they did not explore whether plant genetic diversity increases plant performance by reducing the pressure of plant antagonists or by altering the trophic interactions among plants, herbivores and the natural enemies of herbivores. Perhaps more critical, preceding analyses did not compare genetic diversity effects across different ecosystems (agroecosystems, grasslands, forests, old-field ecosystems, marine ecosystems, wetlands and shrublands), plant life forms (herbaceous vs. woody), experiment types (field plots vs. pot experiments) or climatic zones (tropical and temperate) on a global scale, which is necessary for determining general conditions and contingencies under which any benefits of genetic diversity may be leveraged.

Here, we present a synthesis of 4702 estimates reported in 413 experimental studies that measured the effects of plant genetic diversity on different trophic groups−plants, plant antagonists including invertebrate herbivores, weeds, plant-feeding nematodes and plant diseases (i.e., plant bacteria, fungi and viruses), and natural enemies of herbivores (i.e., predators and parasitoids)−from terrestrial and marine ecosystems around the world (Fig. 1 and Supplementary Data 1). Given that different measures were used to quantify the effects of plant genetic diversity on different trophic groups across studies, we calculated aggregate performance indicators for each trophic group (i.e., plants, plant antagonists, invertebrate herbivores, weeds, plant-feeding nematodes, plant diseases and natural enemies of herbivores) according to Wan et al.[23]. Specifically, these included measures of (i) plant growth, reproduction and quality for plant performance; (ii) herbivore abundance, damage and diversity for invertebrate herbivore performance; (iii) weed growth and diversity for weed performance; (iv) nematode abundance for plant-feeding nematode performance; (v) disease spread and disease damage to plants for plant disease performance; (vi) predator and parasitoid abundance and diversity for natural enemy performance; and (vii) aggregate indicators including the performance of herbivores, weeds, plant-feeding nematodes and plant diseases for plant antagonist performance. The analyses were then carried out considering both the aggregate performance indicator (e.g., invertebrate herbivore performance) and its specific components (e.g., herbivore abundance or herbivore diversity).

We calculated the standardised mean difference between measures of these trophic groups in genetically diverse and monogenotypic

plant stands, used a meta-regression model to analyse responses of trophic groups to plant genetic diversity, and then used piecewise structural equation to analyse trophic interactions using path analysis. We hypothesised that (i) plant genetic diversity increases plant performance directly by an increased complementarity or decreased intensity of plant competition among different plant genotypes, (ii) plant genetic diversity increases plant performance indirectly by reducing the performance of plant antagonists and increasing the performance of natural enemies of herbivores, and (iii) the effects of plant genetic diversity on these multiple trophic levels and their trophic interactions will vary across different plant life forms, ecosystems and climatic zones. Our findings could help future studies to identify the mechanisms by which plant genetic diversity influences plant populations both directly and indirectly.

## Results and discussion
### Effects of plant genetic diversity on multiple trophic groups
We found that plant genetic diversity (i.e. diversification of cropping or plant cultivation systems; see Methods and Supplementary Table 15) decreased the overall performance of plant antagonists (effect size = −0.539, $t = -2.070$, $P = 0.039$) and several of its components (i.e., herbivores (effect size = −0.606, $t = -4.127$, $P < 0.001$), weeds (effect size = −0.071, $t = -0.167$, $P = 0.867$), plant-feeding nematodes (effect size = −2.118, $t = -1.313$, $P = 0.189$) and plant diseases (effect size = −1.087, $t = -5.826$, $P < 0.001$)), while increasing the performance of plants (effect size= 0.344, $t = 9.098$, $P < 0.001$) and natural enemies of herbivores (effect size= 0.778, $t = 4.220$, $P < 0.001$; Fig. 1b). A similar pattern was found when trophic groups were divided into subgroups (e.g., plant performance into plant growth, plant quality and plant reproduction, weed performance into weed growth and weed diversity, and plant disease performance into disease spread and disease damage; see Supplementary Table 3). In the case of small sample sizes (Supplementary Table 3), the results for subgroups were inconclusive (e.g., for herbivore diversity ($N = 9$, effect size= 0.292, $t = 0.991$, $P = 0.322$) and parasitism ($N = 16$, effect size= 0.089, $t = 0.169$, $P = 0.866$)).

In a second step, we tested whether these effects differed among ecosystems, plant life forms, experiment types or climatic zones. In agroecosystems or grasslands, the overall pattern described above was consistently found (Fig. 2a, b; Supplementary Table 4). For the other ecosystems, the responses were variable (Fig. 2c–g; Supplementary Table 4). The overall pattern was also consistently found for both types of experimental studies (i.e., plot and pot experiments) (Fig. 3; Supplementary Table 5), and for both plant life-forms (i.e., herbaceous and woody plants) (Fig. 4; Supplementary Table 6). Across climatic zones, we found a stronger response in temperate than in tropical zones (Fig. 5; Supplementary Table 7), likely because of a smaller sample size in tropical systems (e.g., natural enemies: $N = 19$, effect size= 0.206, $t = 0.388$, $P = 0.698$; weeds: $N = 5$, effect size = −0.290, $t = -0.745$, $P = 0.456$).

At the ecosystem level, we found that plant genetic diversity showed a positive effect on plant performance in agroecosystems (effect size= 0.362, $t = 8.275$, $P < 0.001$), grasslands (effect size= 0.353, $t = 4.670$, $P < 0.001$), old-field ecosystems (effect size= 0.681, $t = 5.573$, $P < 0.001$) and marine ecosystems (effect size= 0.778, $t = 3.753$, $P < 0.001$) (Supplementary Table 4). Two plausible mechanisms could explain the positive effect of plant genetic diversity on plant performance. Firstly, an increased complementarity (i.e., niche partitioning or facilitation) or decreased intensity of plant competition among different plant genotypes[8,9]. Secondly, an increased net positive interactions with higher trophic levels (i.e., increasing genotypic polycultures resulted in a decreased herbivore abundance) that might amplify plant performance[9]. However, the positive effect of plant genetic diversity on plant performance was not consistently found in forests, wetlands or shrublands (Supplementary Table 4). This may be

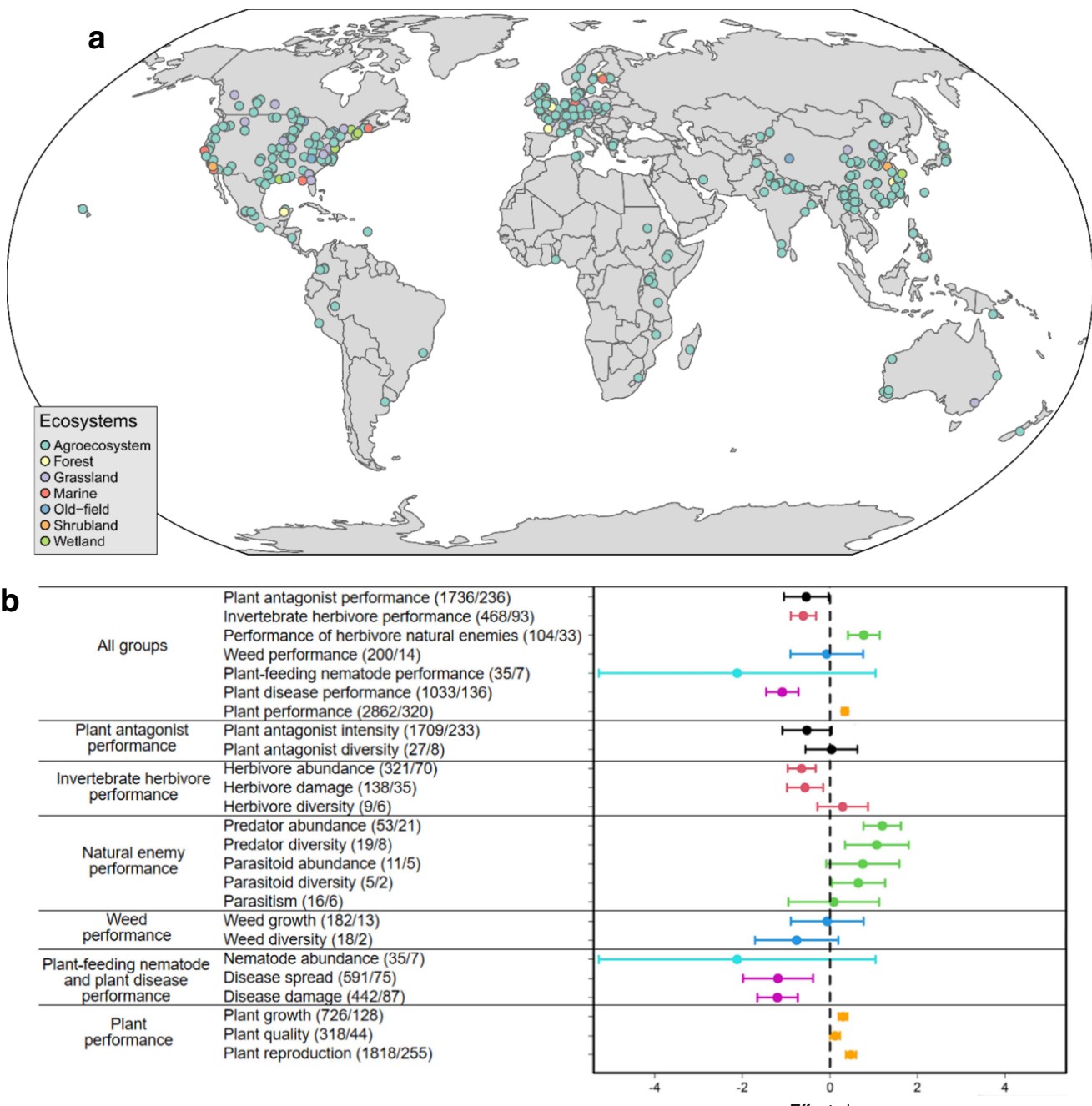

**Fig. 1 | Global distribution of plant genetic diversity study locations and the responses of trophic groups to plant genetic diversity. a** Study locations across all ecosystems (world map in World Robinson projection). **b** Responses of trophic groups across all studies. A literature search identified study locations for agroecosystems (414), forests (16), grasslands (24), old-field ecosystems (14), marine ecosystems (11), wetlands (9) and shrublands (5), respectively, from a total of 413 published articles. Fourty-nine articles included more than one study location (range 2–8). In Fig. 1b, horizontal lines indicate the 95% confidence intervals around the means; numbers in brackets indicate the numbers of observations and studies; seven lines represent plant antagonist (black), invertebrate herbivore (red), natural enemy of invertebrate herbivores (green), weed (blue), plant-feeding nematode (turquoise), plant disease (purple) and plant (orange) performance responses, respectively; plant antagonists include herbivores, weeds, nematodes and diseases; and natural enemies include predators and parasitoids.

due to one or more of the following potential explanations: (i) fewer studies have been conducted in these ecosystems, (ii) at higher genotypic richness, genotype-by-genotype interactions resulted in lower relative performance of each genotype relative to the monoculture yield (i.e., trait-dependent complementarity became more negative at higher genotypic richness treatments)[35], or (iii) plant genetic diversity indirectly decreased plant growth by increasing the abundance and species richness of herbivores[7], as individual genotypes varied in their resistance and susceptibility to herbivory[36,37].

Interestingly, plant genetic diversity decreased herbivore performance in agroecosystems (effect size = −1.008, $t = −5.419$, $P < 0.001$), grasslands (effect size = −0.961, $t = −2.322$, $P = 0.026$), forests (effect size = −0.151, $t = −0.679$, $P = 0.497$) and marine ecosystems (effect size = −0.289, $t = −1.765$, $P = 0.078$) (Supplementary Table 4). This reduction might reflect resource heterogeneity effects on foraging behaviour of herbivores[28], as well as on herbivore movement[38]. Decreased herbivore performance under higher genetic diversity could also be explained by associational resistance in genotypic

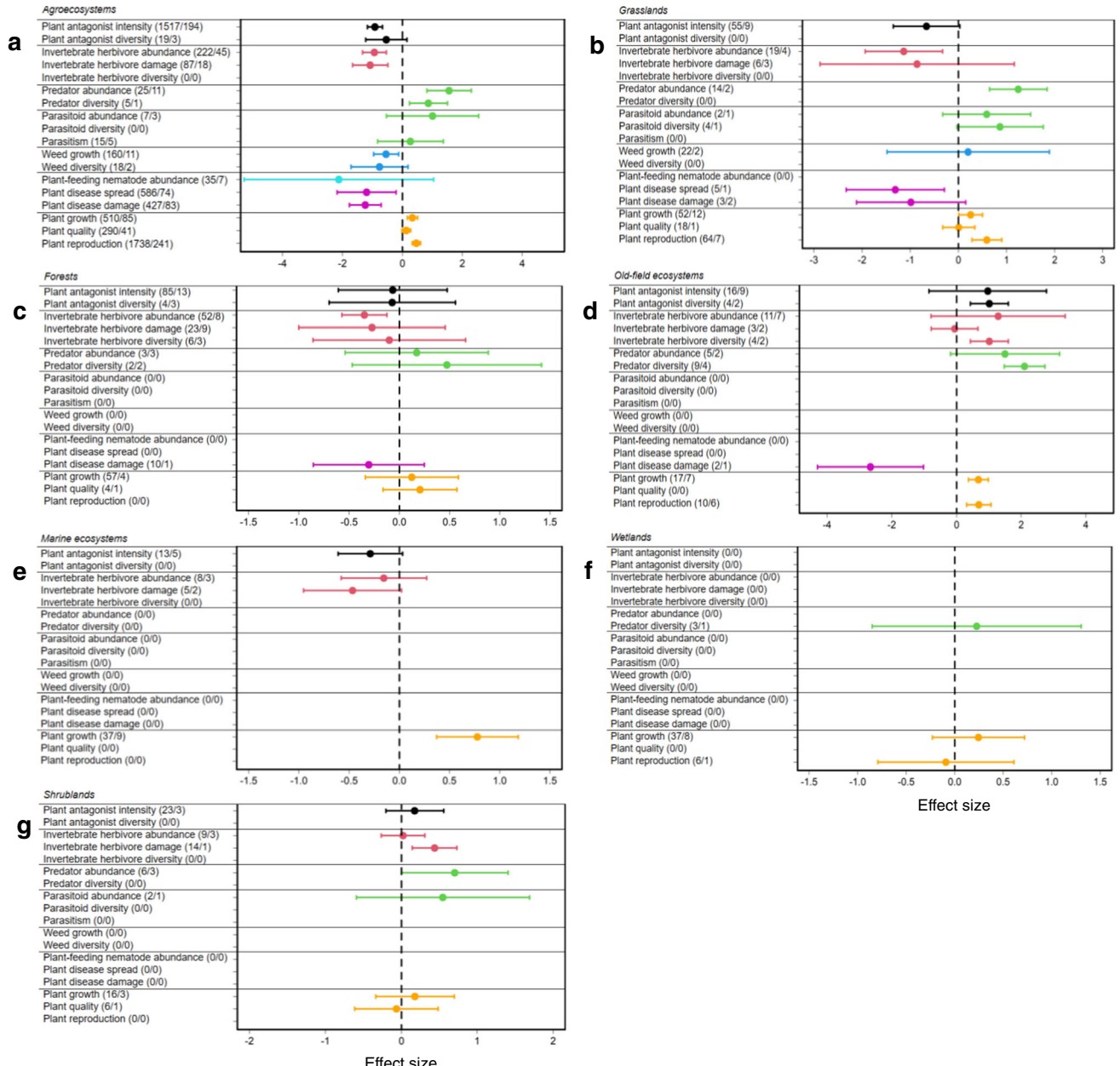

**Fig. 2 | Mean effect sizes of 18 response categories for the seven trophic groups.** **a** Agroecosystems (335 studies). **b** Grasslands (25 studies). **c** Forests (15 studies). **d** Old-field ecosystems (15 studies). **e** Marine ecosystems (11 studies). **f** Wetlands (10 studies). **g** Shrublands (6 studies). Horizontal lines indicate the 95% confidence intervals around the means. Numbers in brackets indicate the numbers of observations and studies. Black, red, green, blue, turquoise, purple and orange lines denote plant antagonists, invertebrate herbivores, natural enemies of herbivores, weeds, plant-feeding nematodes, plant diseases and plants, respectively.

mixtures[4], as many studies focused on control of a single herbivore species by mixing crop or plant genotypes with known differences in resistance to this herbivore. Alternatively, the lower insect herbivore performance under higher genetic diversity may be explained by the resource concentration hypothesis (RCH). Although RCH has been generally tested considering plant species diversity, the concept may be extended to plant genetic diversity, as herbivores have been shown to be able to distinguish between plant genotypes[19]. In addition, an increase in plant volatiles or plant secondary metabolites from the wide range of genetic and chemical diversity within plant species[39,40], might contribute to the release of defensive chemicals to control or repel herbivores[41], and thus result in a decreased herbivore performance in plant genotypic mixtures.

On the other hand, plant genetic diversity was associated with an increase in herbivore performance in old-field (effect size=1.102,

$t = 1.224$, $P = 0.221$) and shrub systems (effect size=0.178, $t = 0.912$, $P = 0.362$). In this case, complementarity in resource use among plant genotypes might have increased plant growth and quality, resulting in increased herbivore abundance[9,42,43]. Similarly, herbivores may be attracted by plant volatiles from plant genetic diversity[42,43]. Yet, the increased herbivore abundance could also have been driven by associational susceptibility in genotypically diverse plots where the ramets of otherwise resistant genotypes could be attacked by herbivores due to their close proximity to susceptible genotypes[44,45].

We also found that plant genetic diversity reduced (i) weeds in agroecosystems (effect size = −0.582, $t = −3.350$, $P = 0.001$; Fig. 2a, b); (ii) diseases in agroecosystems (effect size = −1.085, $t = −5.211$, $P < 0.001$), grasslands (effect size = −1.161, $t = −3.031$, $P = 0.002$) and old-field ecosystems (effect size = −2.659, $t = −3.176$, $P = 0.002$) (Fig. 2a–d), and (iii) plant-feeding nematodes in agroecosystems

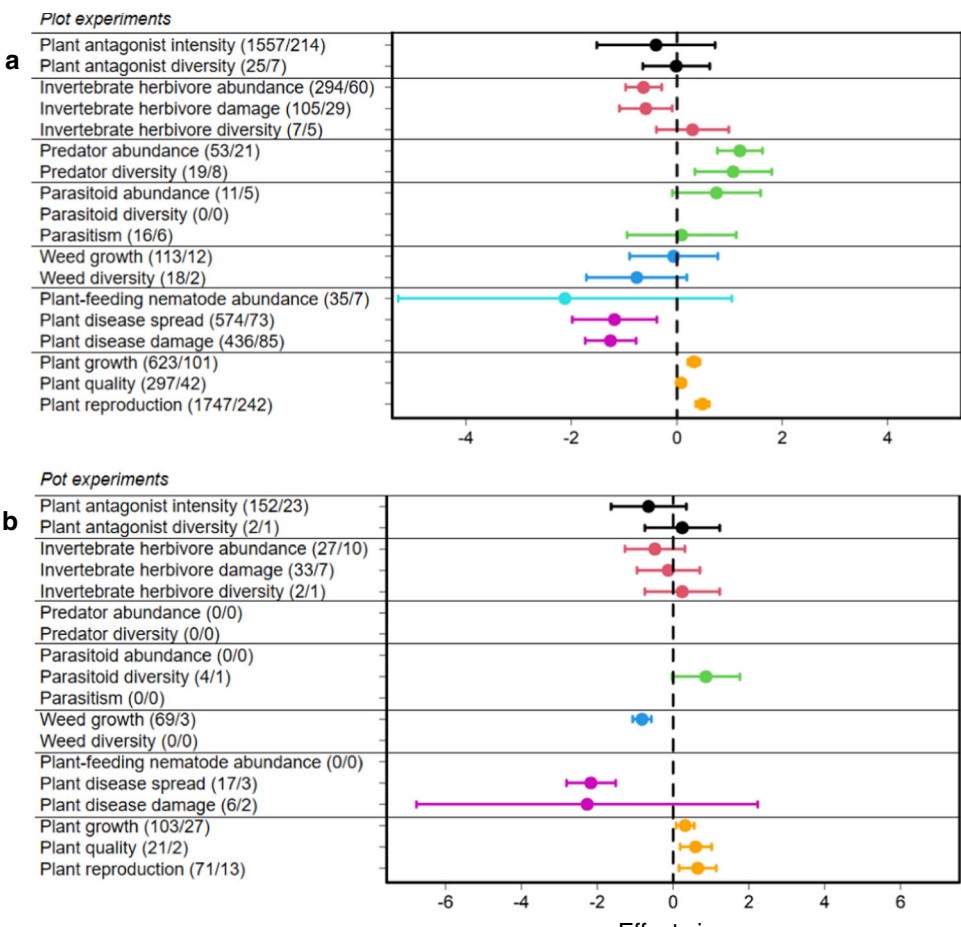

**Fig. 3 | Mean effect sizes of 18 response categories for the seven trophic groups. a** Plot experiments. **b** Pot experiments. Horizontal lines indicate the 95% confidence intervals around the means. Numbers in brackets indicate the numbers of observations and studies. Black, red, green, blue, turquoise, purple and orange lines denote plant antagonists, invertebrate herbivores, natural enemies of herbivores, weeds, plant-feeding nematodes, plant diseases and plants, respectively.

(effect size = −2.118, $t = −1.313$, $P = 0.189$; Fig. 2a), indicating strong biocontrol services. Such biological control effects may be an indication of genetic heterogeneity: (i) inhibiting the growth of weeds through allelopathic effects, detrimental plant secondary metabolites and growth competition[12], (ii) diluting the concentration of resources or disrupting the movement of pathogens between host plants[1], or (iii) creating rhizosphere inhibition zones against nematodes[46]. Generally, physical barriers and variety resistance may account for the effects of plant genetic diversity on decreases in insect herbivores, nematodes or diseases (e.g., altered dispersal and transmission rates of air-borne pathogens, splash-borne propagules or soil-borne bacteria)[47].

Plant genetic diversity also had a direct influence on higher trophic levels across ecosystems. Specifically, we found that more genetically diverse plant stands supported more natural enemies of herbivores, such as predators and parasitoids (Fig. 1b). Such effects can be direct[15,32], mediated by increases in herbivore abundance[31,42], driven by trait-mediated indirect effects[48,49], or directly mediated by natural enemies of herbivores through top-down effects as indicated by the enemy hypothesis (EH)[4,50]. According to the EH hypothesis, increasing plant genetic diversity results in a greater 'resource pool' for natural enemies of herbivores, which favours a greater abundance and diversity of predators and parasitoids, and this ultimately leads to stronger enemy top-down effects on herbivore populations. In addition, an increased diversity or amount of plant volatiles at higher plant genetic diversity can also attract more natural enemies to plants[51]. Such top-down effects may partly explain a positive influence of plant genetic diversity on plant performance. However, such top-down effects were

not significant in forests (effect size=0.134, $t = 0.602$, $P = 0.548$) where genetically rich communities often grew more slowly and suffered higher levels of herbivory than genetic monocultures[52]. In addition, we found that tree genetic diversity showed a weaker association with the performance of natural enemies of herbivores and related biocontrol services. As forests are more complex systems compared to some other systems (e.g. agroecosystems), it is possible that even low-diversity forests could provide niches for many predator and parasitoid species[53].

To assess whether the level of plant genetic diversity (i.e., the number of added genotypes, relative to the monogenetic control; see Methods and Supplementary Table 15) correlated with effect sizes, we set up generalised least-squares models with the fitted effect sizes as the response variable, and the log-transformed number of added genotypes in the plant genetic diversity treatment over the control as an explanatory variable. The average effect sizes of number of added genotypes on plant antagonist performance were all less than zero (i.e., negative effect size) and meanwhile the effect sizes significantly increased with the number of added genotypes across all studies (d.f. = 1734, $t = 6.657$, $P < 0.001$; Supplementary Fig. 1a) and for agroecosystems (d.f.=1534, $t = 3.472$, $P = 0.001$) or grasslands (d.f. = 53, $t = 2.168$, $P = 0.035$) (Supplementary Figs. 2a, 3a), but significantly decreased for forests (d.f. = 87, $t = −2.040$, $P = 0.044$; Supplementary Fig. 4a). This implies that adding only one plant genotype or more than one plant genotype can potentially inhibit plant antagonists in agroecosystems, grasslands and forests (because the average effect sizes of plant antagonists were all negative, no matter whether the trends of the

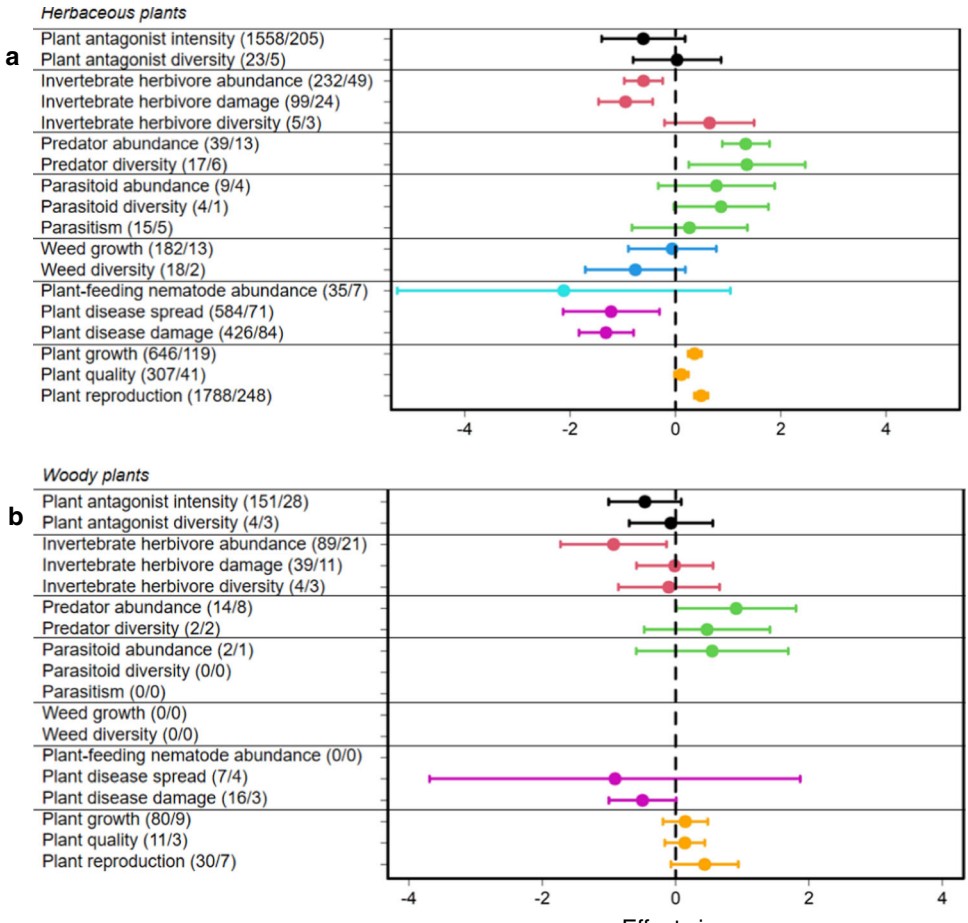

**Fig. 4 | Mean effect sizes of 18 response categories for the seven trophic groups. a** Herbaceous plants. **b** Woody plants. Numbers in brackets indicate the numbers of observations and studies. Horizontal lines indicate the 95% confidence intervals around the means. Black, red, green, blue, turquoise, purple and orange lines denote plant antagonists, invertebrate herbivores, natural enemies of herbivores, weeds, plant-feeding nematodes, plant diseases and plants, respectively.

meta-regression lines for plant antagonists increased in agroecosystems and grasslands or decreased in forests). However, these inhibitory effects on plant antagonists were diminished with the number of added plant genotypes in agroecosystems and grasslands but were enhanced with the number of added plant genotypes in forests (because the trends of the meta-regression lines for plant antagonists increased in agroecosystems and grasslands but decreased in forests, although the average effect sizes of plant antagonists were all negative in these three ecosystems). Weakened inhibitory effects on plant antagonists in agroecosystems and grasslands might be due to the fact that an increased complementarity in plant resource use, as resulting from the increase in the number of added plant genotypes[9,42,43], may benefit plant growth (Supplementary Figs. 1g, 4g) and thus gives rise to a decrease in inhibitory effect. For example, compared with monoculture controls, the abundance of the invertebrate herbivores was lower in treatments with multiple plant genotypes and meanwhile such herbivore abundance increased with the increase in plant genotypes.

In agroecosystems[12,38], grasslands[49] or forests[30,31], for example, intercropping and cover vegetation are commonly applied and the number of added plant genotypes used is often smaller (2–4 in general). Thus, enhanced pest control can be realised by adding another one-to-three plant genotypes in agroecosystems, grasslands or forests. However, there were no significant differences between adding one and adding more than one plant genotypes in old-field ecosystems (d.f.=18, $t = 0.774$, $P = 0.449$), marine ecosystems (d.f.=11, $t = 0.115$, $P = 0.289$) and shrublands (d.f.=21, $t = 0.985$, $P = 0.336$) (Supplementary Figs. 5, 6 and 8), which might be an artefact of fewer studies

documenting plant antagonist responses to increased plant diversity in old-field ecosystems ($N = 20$), marine ecosystems ($N = 13$) and shrublands ($N = 23$). We found significantly positive relationships between plant antagonists and the number of added genotypes in plot experiments (d.f.=1580, $t = 5.265$, $P < 0.001$), pot experiments (d.f.=152, $t = 2.121$, $P = 0.036$), herbaceous plants (d.f. = 1579, $t = 6.133$, $P < 0.001$), woody plants (d.f.=153, $t = 2.558$, $P = 0.012$) or temperate zones (d.f. = 1488, $t = 5.512$, $P < 0.001$) (Supplementary Figs. 9–13), but no significant relationships in tropical ecosystems (d.f. = 138, $t = 1.565$, $P = 0.120$) (Supplementary Fig. 14). Overall, there were variable relationships between the number of added genotypes and the performance of herbivores, their natural enemies, weeds, nematodes, plant diseases or plants for individual ecosystem types, experimental study types, plant life forms and climatic zone types (Supplementary Figs. 1–14; Supplementary Table 16).

## Effects of plant genetic diversity on trophic interactions

We obtained 1606 estimates of interactions between pairs of trophic levels derived from 163 studies testing the effects of plant genetic diversity across multiple trophic levels. First, we tested the effect of plant genetic diversity on bi-trophic interactions between plants and plant antagonists (1484 estimates derived from 139 studies) using multilevel piecewise structural equation models. In these models, different plant antagonists (i.e., invertebrate herbivores, weeds, plant-feeding nematodes or plant diseases) were considered together. We found that plant genetic diversity affected plant performance directly and indirectly by reducing the performance of plant antagonists

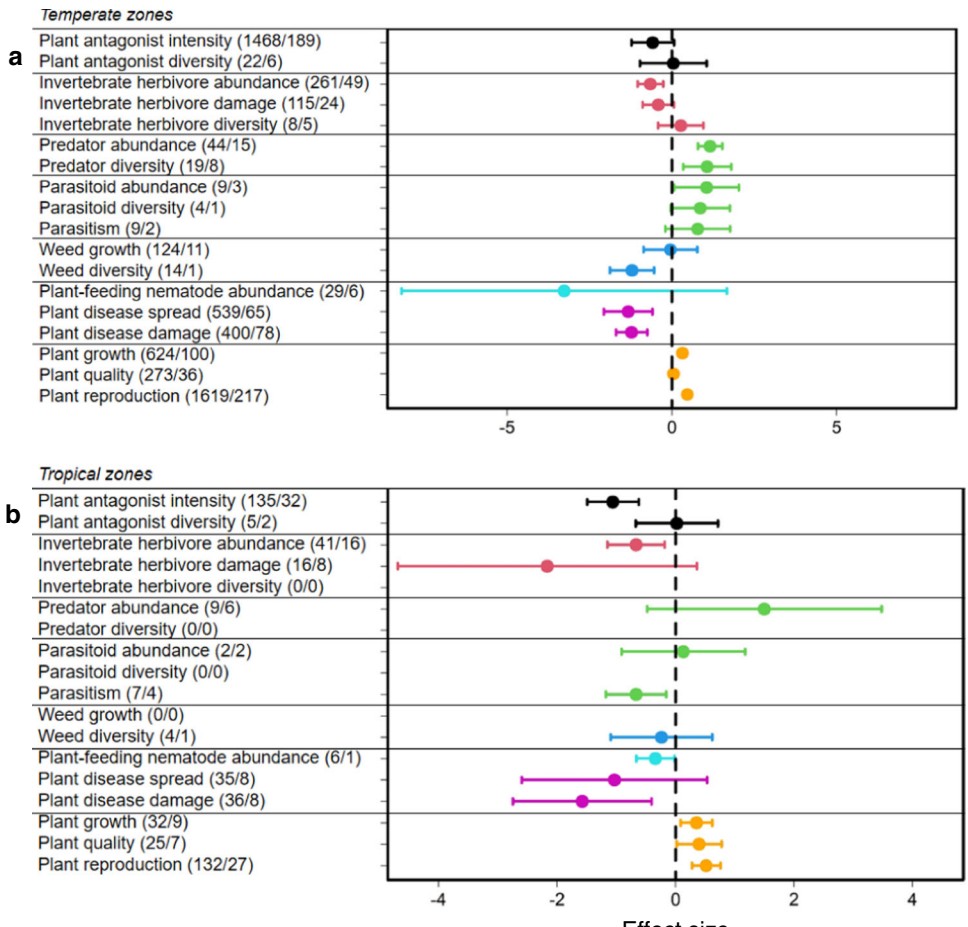

**Fig. 5 | Mean effect sizes of 18 response categories for the seven trophic groups. a** In temperate zones. **b** In tropical zones. Horizontal lines indicate the 95% confidence intervals around the means. Numbers in brackets indicate the numbers of observations and studies. Black, red, green, blue, turquoise, purple and orange lines denote plant antagonists, invertebrate herbivores, natural enemies of herbivores, weeds, plant-feeding nematodes, plant diseases and plants, respectively. Data from greenhouse and other indoor experiments are not shown in this figure.

(Fig. 6). The same pattern was also consistently found in agroecosystems, but not in other ecosystems (Supplementary Fig. 19). In these cases, plant genetic diversity often showed a direct effect on plant or plant antagonist performance, but no indirect effects on plant performance mediated by effects on plant antagonists. Furthermore, we found no evidence of an indirect effect of plant genetic diversity on plant performance through a reduction of plant antagonist pressure when we separately tested herbivore, weed or nematode performance (Supplementary Fig. 18a–c). However, we consistently found a direct effect of plant genetic diversity on the performance of plants and their antagonists. It is likely that further studies will be needed to validate such models. Indeed, when more data were available, as for the case of disease performance ($N = 969$), such mediating effects were evident (Supplementary Fig. 18d). However, when we tested the effects of number of added plant genotypes on the bi-trophic interactions, we found that the direct and indirect effects of number of added plant genotypes on plants, weeds and plant diseases were different from those of plant genetic diversity (Supplementary Fig. 15b–d), and that the direct and indirect effects of number of added plant genotypes on plants and plant antagonists were also different from those of plant genetic diversity across different ecosystems (Supplementary Fig. 16), experiment types, plant life forms and climatic zones (Supplementary Fig. 17), respectively.

For a subset of studies ($N = 91$), where the effect of plant genetic diversity was investigated for all three trophic levels (plant-herbivore-natural enemy interactions), we tested effects of plant genetic diversity on tri-trophic interactions. Structural equation modelling showed a significant direct influence of plant genetic diversity on herbivore (estimate = −0.865, $P = 0.016$) and plant performance (estimate = 0.884, $P = 0.010$), but no significant indirect effects mediated via trophic cascades (plant genetic diversity vs. natural enemies: estimate= 0.709, $P = 0.228$; natural enemies vs. hebivores: estimate = −0.011, $P = 0.906$; herbivores vs. plants: estimate = −0.025, $P = 0.647$). Specifically, we found a positive effect of plant genetic diversity on herbivore natural enemy and plant performances and a negative effect on herbivore performance. These findings indicate a potential effect of plant genetic diversity on a tri-trophic cascade (Fig. 7). However, when we tested the effects of number of added plant genotypes on the tri-trophic interactions of plants, herbivores and their natural enemies, we found that the direct (natural enemies: estimate= 0.069, $P = 0.657$; herbivores: estimate = −0.035, $P = 0.918$; plants: estimate= 0.106, $P = 0.440$) and indirect (natural enemies vs. hebivores: estimate = −0.012, $P = 0.905$; herbivores vs. plants: estimate = −0.037, $P = 0.511$) effects were not significant (Supplementary Table 9; Supplementary Fig. 15a).

Our synthesis comprehensively shows that plant genetic diversity directly increases plant performance in terrestrial and aquatic systems on Earth, which has been partially shown also for agroecosystems in which cultivar mixtures increased the yield of maize, legumes, wheat, oats, barley, soybean and sorghum[18]. Our results also indicate that plant genetic diversity promotes ecosystem services by strengthening trophic interactions: benefiting natural enemies of herbivores

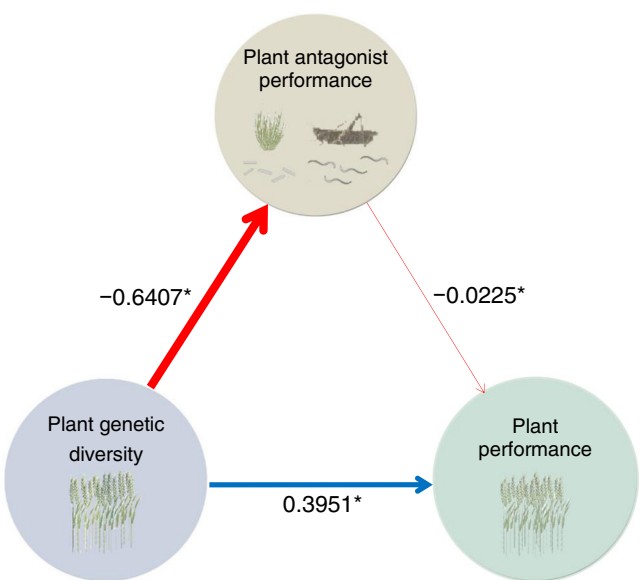

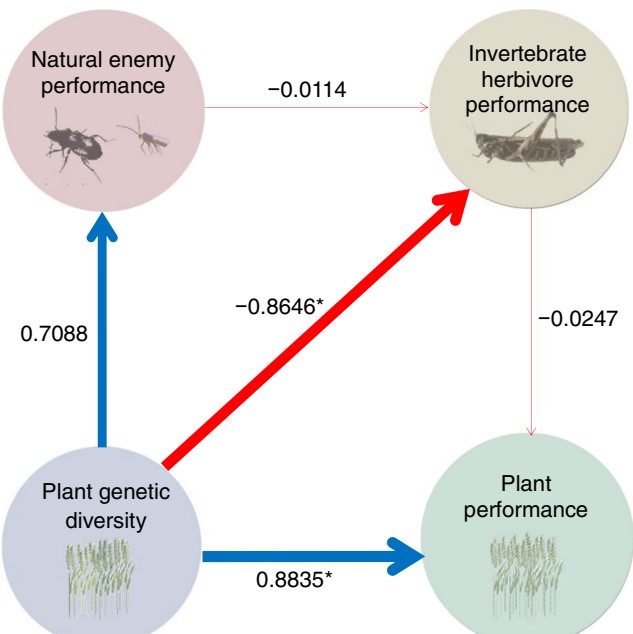

**Fig. 6 | Piecewise structural equation model for the effects of plant genetic diversity on bi-trophic interactions of plant antagonist performance and plant performance across global ecosystems.** The effects of plant genetic diversity (measured as standardised mean difference betweem genetically diverse and genetically monogenotypic plant stands) on bi-trophic interactions of invertebrate herbivore and plant performance, weed and plant performance, plant-feeding nematode and plant performance, and plant disease and plant performance are presented in Supplementary Fig. 18a–d, respectively. Plant genetic diversity is shown in dusty blue. Plant antagonist performance including herbivore performance (abundance, damage and diversity of herbivores), weed performance (growth and diversity of weeds), plant-feeding nematode performance (nematode abundance) and plant disease performance (disease spread and damage) is shown in beige circles. Plant performance (growth, quality and reproduction of plants) is shown in teal colour. Blue and red arrows denote positive and negative relationships, respectively; numbers next to each arrow are the estimated coefficients from piecewise structural equation models, and line width is proportional to the magnitude of the coefficients (Supplementary Tables 10, 11).

**Fig. 7 | Piecewise structural equation model for the effects of plant genetic diversity on tri-trophic interactions of plants, invertebrate herbivores, and the natural enemies of herbivores.** Plant genetic diversity is shown in dusty blue. Natural enemy performance (predator abundance, predator diversity, parasitoid abundance, parasitoid diversity and parasitism) is shown in pink. Herbivore performance (abundance, damage and diversity of herbivores) is shown in beige circles. Plant performance (growth, quality and reproduction of plants) is shown in teal colour. The blue and red arrows denote positive and negative relationships, respectively; numbers next each arrow are the estimated coefficients from piecewise structural equation models, and line width is proportional to the magnitude of the coefficients (Supplementary Tables 8, 9). The asterisks indicate the significance at 5% level.

including invertebrate predators and parasitoids (in line with the conclusion that plant genetic diversity benefited predators from 162 estimates of effect size in 60 experimental studies[21]), in turn suppressing invertebrate herbivores, and enhancing plant performance; while also enhancing plant performance by suppressing weeds, plant-feeding nematodes and diseases. These findings contribute to explaining the mechanisms by which manipulation of plant genetic diversity may affect different trophic groups and their interactions. From an applied perspective, promoting plant genetic diversity may help promote both pest control in managed ecosystems and consumer control of plant production in natural ecosystems via strengthening top-down control, enhancing associated ecosystem functions and services. In conclusion, our results indicate that plant genetic diversity can help society, decision-makers and other stakeholders to take advantage of the important biocontrol services provided by plant genetic diversity on Earth.

## Methods
### Study Selection
We conducted a literature search on the Web of Science and China National Knowledge Internet (www.cnki.net) (last accessed in September 2021) using the Boolean search string: ["plant genetic diversity" OR "plant genotypic diversity" OR "crop genetic diversity" OR "crop genotypic diversity" OR "intraspecific diversity" OR "inter-genotypic" OR "intervarietal" OR "resistant *susceptible cultivar*" OR "pure * mixed cultivar" OR "cultivar mixture" OR "varietal mixture"] AND

["predat*" OR "herbivor*" OR "parasitoid" OR "wasp*" OR "natural enem*" OR "pest management" OR "pest control" OR "biological control" OR "plant disease" OR "plant virus" OR "nematode" OR "weed" OR "yield" OR "productivity" OR "biomass"]. Overall, about 145000 papers were screened for relevance. Additionally, references were obtained from 27 review papers identified in the original search (Supplementary References). Finally, we arrived at 413 papers based on the following criteria: the study included at least one comparison between plant stands with one genotype (monoculture control treatment) or ≥2 genotypes (mixed treatment); the use of pesticides and other practices (fertiliser, irrigation, etc) should be the same for the control and mixed treatments; both the control and the mixed treatment had one and the same plant species; and the measurements of treatment and control groups were performed at the same spatiotemporal scale. Data were extracted from graphs using the "GetData Graph Digitizer" software[23]. We first used the data for which the authors in a cited paper had listed the mean values of multiple sampling dates or multiple sampling years. If the authors did not present these mean values, we adopted the data of the latest sampling date[23] (more details in Supplementary Methods).

When a study covered multiple levels of plant genotypes, measurements of monoculture stands and different numbers of plant genotypes were treated as independent observations. For studies that included more than one location, we considered these experimental observations separately in each location and used the longitudes and latitudes of all locations in Fig. 1, respectively. When the means of observed weed performance indicators (e.g., weed growth in one study), or the indicators for herbivores (e.g., herbivore damage in one study) and predators (e.g., predator abundance in one study) were not given in a study, we extracted these values directly from the figures

(e.g. if a linear or a non-linear relationship between plant genetic diversity and one of these indicators was presented in a figure in the paper, we extracted the values from the fitting equations). When the treatment group was paired with the control group, we excluded multiple comparisons within a single study, and we selected different comparison data (observations with pure or /single plant genetic diversity were considered as control group while the others as the treatment group). Thus, we tried our best to diminish the possibility that some of these 4702 effect sizes might inflate the results with meta-analytical replicates.

## Predictor variables

We tested six categorical variables and one continuous variable as predictors of genetic diversity effect (a detailed description is provided in the Supplementary Methods). (1) Trophic group: a categorical variable that denotes whether the studied organisms were invertebrate herbivores (i.e., arthropod herbivores, amphipod herbivores and molluscan herbivores), natural enemies of invertebrate herbivores (i.e., carnivores of invertebrate herbivores and parasitic wasps), weeds (harmful plants in managed ecosystems dominated by other plants or crops), plant-feeding nematodes, or plant diseases (plant bacteria, fungi and viruses which infest or infect plants and cause damage to plants); in addition, we considered an integrated categorical variable, i.e. plant antagonists[54], which included invertebrate herbivores, weeds, plant-feeding nematodes and plant diseases; (2) Response variable type: abundance and diversity of herbivores as well as herbivory damage; abundance and diversity of invertebrate predators; abundance, diversity and parasitism of parasitoids; growth and diversity of weeds; nematode abundance; disease spread and disease damage; growth, reproduction and quality of plants; "diversity" of herbivores, predators or parasitoids, respectively, encompasses both species richness and Shannon Diversity; (3) Ecosystem type: agroecosystems, old-field ecosystems, marine ecosystems, grasslands, forests, shrublands or wetlands (number of cited studies for each ecosystem type included in this paper should be more than 3, as presented in Supplementary Data 1)[23]; (4) Plant life form: herbaceous or woody plants;[23] (5) Experiment type: plot or pot experiments; specially, plot experiments were done in terrestrial field ecosystems, common garden experiments with a few or several replicated plots, or aquatic ecosystems with a few or several replicated plots, whereas tray, box, tanker and container experiments were considered as pot experiments; (6) Climatic zone type: temperate or tropical (data from greenhouse and indoor experiments were removed from models including the climatic zone variable); and (7) Number of added plant genotypes: a continuous variable presenting the number of genotypes by which the target plant stand was enhanced in the diversity treatment (≥2 genotypes) compared to the single genotype control (specifically, through methods such as interplanting, undersowing, intercropping, mixed cropping and mixed planting).

## Definition of plant genetic diversity

In this manuscript, we refer to "plant genetic diversity" as any activity that increases genetic diversity in a plant community, relative to a monospecific control treatment. Thus, this is essentially a binary variable (zero or one), indicating whether plant genetic diversity had been increased, but irrespective of the number of genotypes added. Results are for example shown in the path analyses for Figs. 6, 7 and for Supplementary Figs. 18–20.

## Definition of effect size and its measures

In the context of the present study, the effect size is defined for each study or observation, as the sign and magnitude of parameter estimates from the individual studies or observations. In addition, when relationships between one variable and another were considered, effect sizes were also represented by the differences in the response variables between levels of independent variables (or between "control" and "experimental" levels) or by raw regression slopes between the response variables and the continuous predictors.

We calculated the Standardised Mean Difference to quantify the plant genetic diversity effects on the various trophic groups as $SMD_i = (M_{ti} - M_{ci})/sd_i, 1 \le, i \le m$, in which $M_{ti}$ and $M_{ci}$ were the mean values in the treatment and control groups, respectively, and $sd_i = \sqrt{\frac{(n_{ti}-1) \times sd_{ti} + (n_{ci}-1) \times sd_{ci}}{n_{ti} + n_{ci} - 2}}$, in which $n_{ti}$ and $n_{ci}$ were the sample sizes and $sd_{ti}$ and $sd_{ci}$ were the standard deviations in the two groups, respectively. An unbiased approach was used to estimate the sampling variance[55]. We used SMD as the response variable for different models throughout the text.

## Meta-regression

Meta-regression[56] was employed to assess whether the effects of plant genetic diversity on different trophic groups could be explained by the various predictor variables and their interactions. Specifically, we fitted multilevel mixed-effects meta-regression models, using the R package metafor (version 3.4-0). The effect size metric SMD was calculated with the function "escalc()" (argument "measure" set to "SMD"), and unbiased sample variance estimates were constructed with the function "escalc()" (argument "vtype" set to "UB"). While the signs of SMDs in trophic groups were the original values for fitting the models, we did not transform the signs of reported meta-estimates for trophic values. Trophic groups and integrated trophic groups (i.e., plants, plant antagonists, natural enemies of herbivores) were included as fixed effects along with the moderator variables (i.e. ecosystem type, type of experimental study, plant life form, climatic zone, and number of added plant genotypes).

The potential effect of plant species was accounted for in all models by 1) adding random intercepts for plant species identity[57,58], and 2) including the phylogenetic relatedness between plant species as part of the correlation structure[59,60]. Furthermore, each effect size was nested within the corresponding study to incorporate the hierarchical error structure of multiple effects coming from the same study. Subsets of the included studies were analysed to better understand the effects within different subgroups (see Supplementary Methods).

For each mixed-effects meta-analysis model, we first fitted a base model by treating the trophic group as the only fixed effect term. Second, the interactions between the trophic group and other predictor variables were also included in the model to assess whether model fit was improved, using a likelihood-ratio test (LRT). Third, the trophic group response category (nested within trophic group) and the interactive effects between the response category and predictors (types of ecosystem, experimental study, plant life form, climatic zone) were also included in the model (using a likelihood-ratio test (LRT) to allow model comparisons) (Supplementary Table 1). For example, the model with "trophic group + ecosystem type" was compared to the base model with just trophic group, and the model with "trophic group + ecosystem type + trophic group × ecosystem type" would be compared to a model with "trophic group + ecosystem type", and these were actually always single-term deletions (Supplementary Tables 1, 2). To examine whether the mean effect sizes in the different categories differed significantly from zero, we acquired t-values with their 95% confidence intervals, which were derived from the fitted meta-regression models.

## Regression analysis of the number of added plant genotypes

To explore if the number of added plant genotypes had additional explanatory power, we ran further analyses using the logarithm of the number of plant genotypes added (e.g., $\log_2(1, 2, 3, 4...)$). We then explored the relationships between the number of added genotypes and the different effect sizes along with fitted meta-regression lines, as presented in Extended Fig. 5–18. These analyses take the number of added genotypes as predictor and SMD as response to display the

trend of SMD with the number of added genotypes. The "gls()" function in the"nlme" library in R[61] was used to fit a linear model using generalised least squares. The errors were allowed to be correlated or have unequal variances, as implemented using an exponential variance function setting "weights=varExp()" in the call to gls. We then employed the R function "Effect()" in the "effects" library[62] to calculate predicted values from the model object and a 95% confidence interval of the predictor considering unequal variances among observations with increasing values of the fitted values.

### Analysis of trophic interactions

We studied how changes in plant genetic diversity impacted the bi-trophic interactions (Fig. 6; Supplementary Figs. 18–20) or tri-trophic interactions (Fig. 7). Furthermore, the $\log_2$-transformed number of added plant genotypes over the control was considered as an extra measurement of plant genetic diversity to explore the magnitude of effect on the trophic interaction relationship (Supplementary Figs. 15–17).

To investigate the bi-trophic and tri-trophic interactions among various trophic groups, we constructed a new data subset containing paired trophic observations (e.g., natural enemy performance vs. invertebrate herbivore performance vs. plant performance, herbivore performance vs. plant performance, weed performance vs. plant performance, plant-feeding nematode performance vs. plant performance, and plant disease performance vs. plant performance; Fig. 7). Then, another data subset was established which encompassed the paired observations of plant antagonist performance vs. plant performance within different ecosystems (i.e., global ecosystems, agroecosystem, grassland, forest, old-field ecosystem, marine ecosystem, wetlands and shrubland; Supplementary Fig. 19). Three additional data subsets were then made, comprising the paired observations of plant antagonist performance vs. plant performance within different types of experimental study, plant life form and climatic zone, respectively (Supplementary Fig. 20). Subsequently, the effect sizes of the responses for each performance (SMD as the response variable) to the predictor variables (i) number of added plant genotypes (Supplementary Figs. 15–17) and (ii) plant genetic diversity (Supplementary Figs. 18–20) were calculated on each data subset, respectively, which were also employed using the R function "factor ()" for conducting the factor analyses (Supplementary Table 15). Additionally, the number of added plant genotypes treated as an external predictor was also included in the factor analyses, using the R function " factor ()" to reveal the effect of the specific value of added plant genotypes on the trophic interactions.

Piecewise structural equation models[63] were fitted on each of the additional data subsets to test the direct and indirect effects of plant genetic diversity and number of added plant genotypes on all potential interactions among the trophic groups, respectively. Briefly, piecewise SEM models were fitted with the R function "psem" in the package "piecewiseSEM", using series of linear mixed-effects models with random intercepts for plots, sites and plant species nested within study IDs. Heteroscedasticity was accounted for by supplying fixed variances based on SMD and setting sigma to 1 in the lme call.

Specifically, we tested the effects of plant genetic diversity on the tri-trophic interactions of herbivore, natural enemy and plant performances, and on each of the bi-trophic interactions between performance values of herbivores and plants, weeds and plants, plant-feeding nematodes and plants, and plant diseases and plants, respectively (Supplementary Fig. 18). Structural equation models were also used to test the effects of plant genetic diversity on the bi-trophic interactions between the performance of all combined plant antagonists and plants in different ecosystems (i.e., global ecosystems, agroecosystem, grassland, forest, old-field ecosystem, marine ecosystem, wetlands and shrubland) (Supplementary Fig. 19), as well as the effects of plant genetic diversity on the bi-trophic interactions

between the combined plant antagonist performance and plant performance within different experiment types (i.e., plot and pot experiments), plant life forms (herbaceous and woody plants) and climatic zones (temperate and tropical zones), respectively (Supplementary Fig. 20). Concurrently, in order to explore the effects of varying levels of genotype diversity, we carried out structural equation modelling to examine the direct and indirect effects of the number of added plant genotypes on the bi-trophic interactions among different pairs of trophic groups (Supplementary Fig. 15), on the bi-trophic interactions of plant antagonist performance and plant performance for different ecosystems (Supplementary Fig. 16), experiment types, plant life forms and climatic zones, respectively (Supplementary Fig. 17). Different meta-regression models were constructed to detect correlations between natural enemy performance and herbivore performance, between herbivore performance and plant performance, and between natural enemy performance and plant performance, respectively, in which the trophic group not being tested was included as a covariate (Supplementary Methods).

### Publication bias test

Publication bias was evaluated by performing regression tests[64] (see Supplementary Table 2). The regression test value (partial slope) tests for an association between effect size and the sample variance, and a significant $P$ value indicates publication bias. For examining the sensitivity of the results, and because the trim-and-fill method[65] could only be used in the context of the fixed- or random-effects model (i.e., in models without moderators), the approach suggested by Egger et al. could not be used in our case[64]. Instead, we followed the suggested approach of Nakagawa and Santos[56], where residuals from different established models were used to estimate publication bias in mixed-effects meta-regression analysis. SMD was used as response variable. Trophic groups, ecosystem types, types of experimental study, plant life forms, climatic zone types, and increased plant genetic diversity were used as predictor, respectively, and sampling variances were used as an additional moderator in the mixed-effect model to test the publication bias. In addition, the Rosenthal fail-safe number for the full dataset[66] was assessed, using a fail-safe number of 101836 for the full dataset (Supplementary methods).

R version 4.1.0[67] was applied for all statistical analyses, in which meta-regression and publication bias were estimated using R package "metafor" 3.4-0. Additionally, the R packages "piecewiseSEM"[68] and "nlme"[69] were used for piecewise structural equation modelling. All tests used 0.05 as significance level.

### Reporting summary

Further information on research design is available in the Nature Portfolio Reporting Summary linked to this article.

## Data availability

The raw and processed data used in this study is available in the Supplementary Files (Supplementary Data 1) and is deposited to Zenodo: https://zenodo.org/record/7302775#.Y2tz8MhzmLk.

## Code availability

The code used to analyse data is deposited to Zenodo: https://zenodo.org/record/7307292#.Y2tzsMhzmLl.

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

## Acknowledgements

The authors thank all of the researchers whose data and work have been included in this meta-analysis, and Professor Bo Li, Jihua Wu, Xiaoqi Zhou, Ming Nie and Zhijie Zhang for providing useful help. N.F.W. was supported by the Shanghai Science and Technology Innovation Action Plan from Shanghai Municipal Science and Technology Commission of China (22015821000) and National Ten Thousand Plan-Young Top Talents of China, Y.Q.H. was supported by the National Natural Science Foundation of China (11971117), and L.F. was supported by the National Natural Science Foundation of China (82204063).

## Author contributions

N.F.W. conceived the idea. N.F.W., L.F., and Y.Q.H. collected the data and drafted the article. N.F.W., L.F., Y.Q.H., and C.S. analysed the data. N.F.W., L.F., M.D., Y.Q.H., L.P.K., F.I., and C.S wrote the manuscript. All authors prepared and edited the final drafts.

## Competing interests

The authors declare no competing interests.
