## [Peer Review File · Nature Communications]

Plant genetic diversity affects multiple trophic levels and trophic interactionsREVIEWER COMMENTS

Reviewer #1 (Remarks to the Author):

General comments

This is a strong paper on an interesting topic. It has an impressive scope and notable results and conclusions that applicable to applied and natural systems. I am familiar with meta-analyses, but not well enough to comment on the methods used. The framing of the paper is appropriate, and the conclusions are reasonable and not overstated given the results that are presented. Kudos to the authors on a nice project. My main concerns lie with the writing which is a bit stiff and could benefit from more thorough editing for clarity. I provide some suggestions below, but by no means are these suggestions comprehensive.

Specific comments:

L81-3: This sentence is awkward. Replace "Meanwhile" with "Moreover" and replace "or not" with "or by altering"

L84: insert "preceding" before "meta-analyses"

L110: Use past tense here and throughout: "decreased" rather than "decreases"

L142: replace "which" with "and"

L154-156: There are too many words between the subject and verb. move "are important factors" to earlier in the sentence; put it right after "effects"

L171: This sentence confused me. I am not sure what "the complex habitat properties" are. Perhaps just removing "the" will help clarify the sentence, but perhaps more revisions are necessary to convey the desired point.

L184-187: Revising this sentence to have the verb closer to the subject will help the clarity of this sentence. As written, I needed to read it a few times to understand it.

L192-194: Revise to remove redundant use of "tritrophic"

L200-211: This is a nice succinct conclusion section, but most of these ideas have been previously expressed by many of the papers included in the meta-analysis and cited earlier in the paper. To acknowledge that the ideas being expressed here are not completely novel, authors should cite in this section relevant papers that drew similar conclusions.

Extended Data Fig. 5 and other similar figures. I think it is better to only present a trend line and an equation when the relationship between the variables is statistically significant, an approach taken that I have seen in many journals. I am not sure about the preference of this journal, but for Fig. 5 I recommend removing trend lines and equations from panels C, F, and G—leaving the P-value makes sense.

Reviewer #2 (Remarks to the Author):

This study presents results from a meta-analysis that covers an impressive breadth of studies that address the importance of intraspecific plant genetic diversity for trophic interactions and plant performance. This is a topic of great interest, both for advancing our fundamental understanding of evolutionary ecology and for improving the sustainability of agroecosystems. The authors have assembled an impressive database, including 4702 comparisons between monocultures and mixtures taken from 413 studies. The database is strongly skewed toward agroecosystems, which is expected based on the availability of past studies in this area, but I appreciate that they also included natural systems in the analysis. Overall, their results show that increasing intraspecific genetic diversity in plant populations leads to reduced performance of plant antagonists, increased performance of 3rd trophic level natural enemies, and increased performance of plants. For agroecosystems, this result could be an important motivation to increase genetic diversity in crop fields, which could lead to a substantial reduction in pest pressure.

Although I found the main result of the study compelling, I have several major concerns about the manuscript that may limit its impact:

First, the authors present a large number of nuanced results, and the motivation for these analyses was often unclear. There were no hypotheses presented for why the effect sizes should

differ across ecosystem types, plant life forms, or climatic zones, yet this was a major focus of the results. Similarly, the authors do not explain their motivation or any possible mechanisms for the direct and indirect effects of genetic diversity on plant performance (which was the subject of the SEMs). Doing this well will be quite challenging, in large part because they have included so many different types of response variables in their study.

Second, a major concern about the meta-analysis models is that they did not include any phylogenetic controls, or even a random effect of species identity. This is a major concern, especially with a focus on crop plants, because most crops in modern agriculture come from a few key families (Poaceae, Solanaceae, Brassicaceae, etc.) and the literature is almost certainly biased toward some of our most important crops. The species are not listed anywhere that I could find, but many of the comparisons likely are drawn from the same species (e.g. a quick scan of the references suggests that a large number of studies were conducted on wheat), and this is a major source of non-independence (i.e. pseudo-replication) if not properly accounted for in the models.

Finally, even though I am familiar with meta-analyses and have used the R package metafor extensively, I found the methodology and results very difficult to follow. Descriptions of the model structures, response variables, and results were often unclear, making it difficult to evaluate the study.

Other comments:

L33: Suggest rewording to "Intraspecific genetic diversity is...". The parenthetical currently reads as if intraspecific diversity is being defined as genetic diversity.

L 40: consider using "plant antagonists" instead of "plant-damaging organisms", here and throughout the MS

L81: Not sure "Meanwhile" is the right word here

L82: remove "not"

L94: remove parentheses

L91-99: I think it will be very tricky to interpret results with these broad categories of performance. For example, "herbivore performance" includes a huge variety of metrics that could range from the abundance of a single herbivore species that is a major pest to the overall diversity of herbivores present, to the total damage to the plant. In particular, it seems important to distinguish between how plant genetic diversity affects the abundance or pressure from any single herbivore, to how plant genetic diversity affects the herbivore community as a whole (e.g. herbivore diversity). I would expect a more genetically diverse crop field to have lower abundance of any single herbivore species, but a more diverse herbivore community overall.

L91-99: This list would be much easier to read if it were rephrased to use parallel structure across the list elements

L100: monospecific? Or monogenotypic?

L141-146: There seems to be very few studies in the old-field category, and the plant species and herbivores are likely totally different from those in agroecosystems, so I think there are likely many possible mechanisms here, some of which may involve random chance more the biological processes.

L142: rephrase "which potential mechanisms"

L162: Rephrase "In light of EH hypothesis" , maybe "According to the EH hypothesis..."

L165: the idea that more diverse genotypes might benefit natural enemies due to increased host biomass seems counter to the results of the study

L249: "possibility"

L296: monogenotypic?

L300-303: This discussion of the random effects is unclear—including the study ID does not "avoid the heterogeneity" so much as account for non-independence. Also the referenced study (Ref #47) does not seem relevant here?

L 300-303: Another very important source of non-independence among these effect sizes that should be incorporated into the models is due to the fact that multiple studies were conducted on the same species, and many species (especially for crops) are closely related phylogenetically. Thus, species should be included as a random effect with their phylogenetic relatedness as part of the correlation structure. This can be done easily in metafor.

L308-314: It is unclear to me from this description, the supplementary methods, and tables exactly what the model structures were and which models were compared with which using LRTs. The best I could gather (from Supplementary Table 2) is that a model with a single moderator/predictor (e.g. trophic group) was compared to a model that had two moderators and their interaction (e.g. trophic group x plant life form). But this is not a single-term deletion (the interaction and the single effect of plant life form have been removed together), so we cannot assess whether there is an interaction between life form and trophic group, or just a single effect of life form.

L390-404: There is not enough information here and in Supplemental Table 3 to understand how the authors went about evaluating publication bias, even after looking at the referenced paper (Nakagawa and Santos; where several different approaches are discussed). Typical approaches include funnel plots (which would very helpful to add), and regression tests that test for funnel plot asymmetry. In these regression tests a significant p-value in these tests suggests publication bias. The p-values reported in Supp. Table 3 are highly significant, which may suggest strong publication bias (but I am not really sure what was done so this is not possible to evaluate).

Fig. 2: Should the line between SMD plant genetic diversity and plant-damaging organism performance be red? Also, here and other figs: define SMD

Fig. 3: It would be helpful if the widths of the lines were proportional to the strengths of the coefficients, so the reader can easily see which are the strongest relationships.

Extended data Fig. 1: It would be helpful to give the total number of comparisons/studies for each ecosystem type

Extended data Fig. 5-18: I did not see that these figures were ever referenced or discussed in the MS at all. This seems problematic, especially because the results are counterintuitive based on the general hypotheses and broad patterns presented. In particular, as the intraspecific genetic diversity increases, we generally see that performance of plant-damaging organisms becomes more similar in the mixtures relative to monocultures (i.e. the effect size becomes less negative; Fig. 5a). This same general pattern holds for different types of plant antagonists and across different ecosystems, although there is some variability (Fig. 5-18).

Ext. Data Fig. 21: It is stated that arrow widths are proportional to coefficients but they are not—see comment above.

Ext. Data Fig. 22: Natural enemies are mentioned in legend but not shown on figure.

Supplementary Table 1: Please add the plant species to this table, I think this is very important for the reader to get a sense of the breadth of plant species studied.

Supplementary Table 2: Please rephrase description of response variable, this is difficult to follow.

Response to the referee comments

Replies to Reviewer #1:

Comment: General comments. This is a strong paper on an interesting topic. It has an impressive scope and notable results and conclusions that applicable to applied and natural systems. I am familiar with meta-analyses, but not well enough to comment on the methods used. The framing of the paper is appropriate, and the conclusions are reasonable and not overstated given the results that are presented. Kudos to the authors on a nice project. My main concerns lie with the writing which is a bit stiff and could benefit from more thorough editing for clarity. I provide some suggestions below, but by no means are these suggestions comprehensive.

Response: We thank the reviewer for this positive evaluation. We carefully addressed your comments in reviewing the manuscript. We are grateful for your suggestions, which have improved the quality of our study (see this revision).

Comment: Specific comments: L81-83: This sentence is awkward. Replace “Meanwhile” with “Moreover” and replace “or not” with “or by altering”.

Response: Done as suggested (lines 83-84).

Comment: L84: insert “preceding” before “meta-analyses”.

Response: We have inserted “preceding” before “meta-analyses” (line 86).

Comment: L110: Use past tense here and throughout: “decreased” rather than “decreases”.

Response: We have replaced “decreases” with “decreased” and used past tense here and throughout (line 127 and this revision).

Comment: L142: replace “which” with “and”.

Response: We have revised this sentence (lines 174-177).

Comment: L154-156: There are too many words between the subject and verb. move “are important factors” to earlier in the sentence; put it right after “effects”.

Response: Thanks for useful suggestions. We have revised this sentence (lines 189-191).

Comment: L171: This sentence confused me. I am not sure what “the complex habitat properties” are. Perhaps just removing “the” will help clarify the sentence, but perhaps more revisions are necessary to convey the desired point.

Response: We have now revised the text as follows: *“In addition, we found that increased numbers of tree genotypes showed a weaker association with the performance of natural enemies of herbivores and related biocontrol services. As forests are more complex systems compared to some other systems (e.g. agroecosystems), it is possible that even low-diversity forests could provide niches for many predator and parasitoid species⁴⁸”* (lines 203-207).

Comment: L184-187: Revising this sentence to have the verb closer to the subject will help the clarity of this sentence. As written, I needed to read it a few times to understand it.

Response: We have now revised the text as follows: *“Furthermore, we found no evidence of an indirect effect of plant genetic diversity on plant performance through a reduction of plant antagonist pressure when we separately tested herbivore, weed or nematode performance (Extended Data Figs. 22a–c)”* (lines 236-238).

Comment: L192-194: Revise to remove redundant use of “tritrophic”.

Response: We have revised this sentence (lines 243-245).

Comment: L200-211: This is a nice succinct conclusion section, but most of these ideas have been previously expressed by many of the papers included in the meta-analysis and cited earlier in the paper. To acknowledge that the ideas being expressed here are not completely novel, authors should cite in this section relevant papers that drew similar conclusions.

Response: We have cited the two most relevant papers (References 20 and 17) in the conclusion (lines 251-262).

Comment: Extended Data Fig. 5 and other similar figures. I think it is better to only present a trend line and an equation when the relationship between the variables is statistically significant, an approach taken that I have seen in many journals. I am not sure about the preference of this journal, but for Fig. 5 I recommend removing trend lines and equations from panels C, F, and G—leaving the P-value makes sense.

Response: Done as suggested (see Extended Data Figs. 5-18 in this revision).

Replies to Reviewer #2:

Comment: This study presents results from a meta-analysis that covers an impressive breadth of studies that address the importance of intraspecific plant genetic diversity for trophic interactions and plant performance. This is a topic of great interest, both for

advancing our fundamental understanding of evolutionary ecology and for improving the sustainability of agroecosystems. The authors have assembled an impressive database, including 4702 comparisons between monocultures and mixtures taken from 413 studies. The database is strongly skewed toward agroecosystems, which is expected based on the availability of past studies in this area, but I appreciate that they also included natural systems in the analysis. Overall, their results show that increasing intraspecific genetic diversity in plant populations leads to reduced performance of plant antagonists, increased performance of 3rd trophic level natural enemies, and increased performance of plants. For agroecosystems, this result could be an important motivation to increase genetic diversity in crop fields, which could lead to a substantial reduction in pest pressure.

Response: We thank the reviewer for this positive evaluation. We carefully addressed your comments in revising the manuscript. We are grateful for your suggestions, which have improved the quality of our study (see this revision).

Comment: Although I found the main result of the study compelling, I have several major concerns about the manuscript that may limit its impact: First, the authors present a large number of nuanced results, and the motivation for these analyses was often unclear.

Response: We have added and clarified the motivation for these analyses in the Introduction section: **1)** *“these preceding meta-analyses did not compare genetic diversity effects across different ecosystems (agroecosystems, grasslands, forests, old-field ecosystems, marine ecosystems, wetlands and shrublands), plant life forms (herbaceous vs. woody), experiment types (field plots vs. pot experiments) or climatic zones (tropical and temperate) on a global scale, which is necessary for determining the conditions and contingencies under which any benefits of genetic diversity may be leveraged”* (lines 86-91); and **2)** *“Our study thus addresses the mechanisms by which plant genetic diversity influences plants by affecting plant-plant antagonist and plant-herbivore-natural enemy interactions, as well as eco-evolutionary feedbacks stemming from such direct and indirect effects”* (lines 121-124).

Comment: There were no hypotheses presented for why the effect sizes should differ across ecosystem types, plant life forms, or climatic zones, yet this was a major focus of the results.

Response: We have clarified our hypotheses as follows: *“We hypothesized that: (i) plant genetic diversity increases plant performance directly by an increased complementarity or decreased intensity of plant competition among different plant genotypes, (ii) increased natural enemy top-down and decreased plant antagonist top-down effects from plant genetic diversity increase plant performance by reducing the pressure of plant antagonists,*

and (iii) the effects of plant genetic diversity will differ on these multiple trophic levels and their trophic interactions, depending on different plant life forms, ecosystems and climatic zones.” (lines 114-121). In addition, we have clarified it as follows: “these preceding meta-analyses did not compare genetic diversity effects across different ecosystems (agroecosystems, grasslands, forests, old-field ecosystems, marine ecosystems, wetlands and shrublands), plant life forms (herbaceous vs. woody), experiment types (field plots vs. pot experiments) or climatic zones (tropical and temperate) on a global scale, which is necessary for determining the conditions and contingencies under which any benefits of genetic diversity may be leveraged” (lines 86-91).

Comment: Similarly, the authors do not explain their motivation or any possible mechanisms for the direct and indirect effects of genetic diversity on plant performance (which was the subject of the SEMs). Doing this well will be quite challenging, in large part because they have included so many different types of response variables in their study.

Response: We have explained some possible mechanisms for the direct and indirect effects of genetic diversity on plant performance. The added sentences were as follows: **1)** “At the ecosystem level, we found that plant genetic diversity showed a positive effect on plant performance in agroecosystems, grasslands, old-field ecosystems and marine ecosystems (Supplementary Table 5). Two plausible mechanisms could explain the positive effect of plant genetic diversity on plant performance. Firstly, an increased complementarity (i.e., niche partitioning or facilitation) or decreased intensity of plant competition among different plant genotypes^{8,9}. Secondly, an increased net positive interactions with higher trophic levels (i.e., increasing genotypic polycultures resulted in a decreased herbivore abundance) that might amplify plant performance⁹. However, the positive effect of plant genetic diversity on plant performance was not consistently found in forests, wetlands or shrublands (Supplementary Table 5). This may be due to one or more of the following potential explanations: (i) fewer studies have been conducted in these ecosystems, (ii) at higher genotypic richness, genotype-by-genotype interactions resulted in lower relative performance of each genotype relative to the monoculture yield (i.e., trait-dependent complementarity became more negative at higher genotypic richness treatments)³⁴, or (iii) increased plant genetic diversity indirectly decreased plant growth by increasing the abundance and species richness of herbivores⁷, as individual genotypes varied in their resistance and susceptibility to herbivory^{35,36}” (lines 146-162); and **2)** “According to the EH hypothesis, increasing plant genetic diversity results in a greater ‘resource pool’ for natural enemies, which favours a greater abundance and diversity of predators and parasitoids, and this ultimately leads to stronger enemy top-down effects on herbivore populations. Such top-down effects may partly explain a positive influence of plant genetic diversity on plant performance” (lines 197-201).

Comment: Second, a major concern about the meta-analysis models is that they did not include any phylogenetic controls, or even a random effect of species identity. This is a major concern, especially with a focus on crop plants, because most crops in modern agriculture come from a few key families (Poaceae, Solanaceae, Brassicaceae, etc.) and the literature is almost certainly biased toward some of our most important crops. The species are not listed anywhere that I could find, but many of the comparisons likely are drawn from the same species (e.g. a quick scan of the references suggests that a large number of studies were conducted on wheat), and this is a major source of non-independence (i.e. pseudo-replication) if not properly accounted for in the models.

Response: We thank the referee for bringing up this important point. This has prompted us to do a full phylogenetic re-analysis of our data; we set up a phylogeny for the plant species considered in this study, and included the phylogenetic relatedness as a correlation structure to our mixed-effects models (Supplementary Tables 2 and 3 in this revision). Furthermore, each effect size was nested within the corresponding study to incorporate the hierarchical error structure of multiple effects coming from the same study. Subsets of the included studies were analysed to better understand the effects within different subgroups (lines 354-359).

The full R code used is now provided in the Supplementary Materials; briefly, the R coded for mixed model meta-analysis now reads:

```
rma.mv(yi,vi,mods=~factor(Trophic)-1, random = list(~ 1 | Plant.species.new,~  
1|Plant.species.new.p), R=list(Plant.species.new.p=A), data=Total.data), where yi is the  
SMD for each observation, vi is the corresponding variance, and Plant.species.new.p and  
A are obtained from package V.PhyloMaker that was used to derive the phylogeny of the  
species in the dataset:
```

```
species=read.xlsx('plant species information,1)  
mycor=phylo.maker(species) # retrieve the phylogeny for the species using phylo.maker  
mytree <- compute.brlen(mycor$scenario.3) #compute branch lengths  
A <- vcv(mytree, corr=TRUE) #retrieve the variance-covariance matrix  
Total.data1$Plant.species=factor(Total.data1$Plant.species)  
Total.data1$Plant.species.new=Total.data1$Plant.species  
levels(Total.data1$Plant.species.new)=sort(dimnames(A)[[1]])  
Total.data1$Plant.species.new.p=Total.data1$Plant.species.new
```

Therefore, we have now included phylogenetic controls in the meta-analysis models to account for non-independence for giving more convincing results. In a summary, we have clarified it as “*The potential effect of plant species was accounted for in all models by (1) adding random intercepts for plant species identity^{52,53}, and (2) including the phylogenetic relatedness between plant species as part of the correlation structure^{54,55}. Furthermore,*

each effect size was nested within the corresponding study to incorporate the hierarchical error structure of multiple effects coming from the same study. Subsets of the included studies were analysed to better understand the effects within different subgroups (see Supplementary Methods) (lines 354-359). In addition, we also provide a list of all plant species (Supplementary Table 1 in this revision).

Comment: Finally, even though I am familiar with meta-analyses and have used the R package metafor extensively, I found the methodology and results very difficult to follow. Descriptions of the model structures, response variables, and results were often unclear, making it difficult to evaluate the study.

Response: We thank the referee for these useful suggestions. We have revised the descriptions of the methods, especially of the model structures and response variables, and the results to make them clearer for readers (lines 341-445 in the Methods; Supplementary Tables 1-16 and Supplementary Methods).

Comment: Other comments: L33: Suggest rewording to “Intraspecific genetic diversity is...”. The parenthetical currently reads as if intraspecific diversity is being defined as genetic diversity.

Response: Thanks for useful suggestions. We have replaced “Intraspecific diversity (genetic diversity)” with “Intraspecific genetic diversity” (line 32).

Comment: L40: consider using “plant antagonists” instead of “plant-damaging organisms”, here and throughout the MS.

Response: Thanks for useful suggestions! We have replaced “plant-damaging organisms” with “plant antagonists” here and throughout the MS (see line 39 and this revision).

Comment: L81: Not sure “Meanwhile” is the right word here.

Response: We have replaced “Meanwhile” with “Moreover” (line 83).

Comment: L82: remove “not”.

Response: According to the first and second reviewers, we have revised this sentence and replaced “or not” with “or by altering” (line 84).

Comment: L94: remove parentheses.

Response: We have revised this sentence (lines 97-110).

Comment: L91-99: I think it will be very tricky to interpret results with these broad categories of performance. For example, “herbivore performance” includes a huge variety

of metrics that could range from the abundance of a single herbivore species that is a major pest to the overall diversity of herbivores present, to the total damage to the plant. In particular, it seems important to distinguish between how plant genetic diversity affects the abundance or pressure from any single herbivore, to how plant genetic diversity affects the herbivore community as a whole (e.g. herbivore diversity). I would expect a more genetically diverse crop field to have lower abundance of any single herbivore species, but a more diverse herbivore community overall.

Response: We now provide more details on the responses of herbivore abundance, herbivore damage and herbivore diversity to plant genetic diversity in Fig. 1b and Supplementary Table 4. The results indicate that plant genetic diversity led to decreased herbivore abundance and herbivore damage but increased herbivore diversity (Fig. 1b and Supplementary Table 4).

Comment: L91-99: This list would be much easier to read if it were rephrased to use parallel structure across the list elements.

Response: We have revised the sentences (lines 97-110).

Comment: L100: monospecific? Or monogenotypic?.

Response: monogenotypic plant stands (line 112).

Comment: L141-146: There seems to be very few studies in the old-field category, and the plant species and herbivores are likely totally different from those in agroecosystems, so I think there are likely many possible mechanisms here, some of which may involve random chance more than the biological processes.

Response: We agree with the reviewer that the findings in old-field ecosystems are less relevant and less representative for the overall signal. We have changed the text accordingly. We have clarified it as follows: “*On the other hand, plant genetic diversity was associated with an increase in herbivore performance in old-field and shrub systems. In this case, complementarity in resource use among plant genotypes might have increased plant growth and quality, resulting in increased herbivore abundance^{9,38,39}. At the same time, plant genetic diversity could also increase the attraction of herbivores to airborne volatiles^{38,39}. Yet, the increased herbivore abundance could also have been driven by associational susceptibility in genotypically diverse plots where the ramets of otherwise resistant genotypes could be attacked by herbivores due to their close proximity to susceptible genotypes^{40,41}*” (lines 174-181).

Comment: L142: rephrase “which potential mechanisms”.

Response: We have revised the sentences (lines 174-181).

Comment: L162: Rephrase “In light of EH hypothesis”, maybe “According to the EH hypothesis…”.

Response: Done as suggested (line 197).

Comment: L165: the idea that more diverse genotypes might benefit natural enemies due to increased host biomass seems counter to the results of the study.

Response: These sentences were wrongly expressed. We have revised these sentences (lines 197-200).

Comment: L249: “possibility”.

Response: We have replaced “possible” with “possibility” (line 300).

Comment: L296: monogenotypic?.

Response: We have revised this sentence (lines 342-344).

Comment: L300-303: This discussion of the random effects is unclear—including the study ID does not “avoid the heterogeneity” so much as account for non-independence.

Response: In these version, we have clarified it as “*The potential effect of plant species was accounted for in all models by (1) adding random intercepts for plant species identity^{52,53}, and (2) including the phylogenetic relatedness between plant species as part of the correlation structure^{54,55}. Furthermore, each effect size was nested within the corresponding study to incorporate the hierarchical error structure of multiple effects coming from the same study. Subsets of the included studies were analysed to better understand the effects within different subgroups (see Supplementary Methods)*” (lines 354-359).

Comment: Also the referenced study (Ref #47) does not seem relevant here?

Response: We have replaced the referenced study (Ref #47) with the following two studies (lines 354-355):

52. Cardinale, B. J. et al. Impacts of plant diversity on biomass production increase through time because of species complementarity. *Proc. Natl. Acad. Sci. U. S. A.* 104, 18123–18128 (2007).

53. Liu, D., Chang, P. S., Power, S. A. & Bell, J. N. B. Manning P. Changes in plant species abundance alter the multifunctionality and functional space of heathland ecosystems. *New Phytol.* 232, 1238–1249 (2021).

Comment: L300-303: Another very important source of non-independence among these

effect sizes that should be incorporated into the models is due to the fact that multiple studies were conducted on the same species, and many species (especially for crops) are closely related phylogenetically. Thus, species should be included as a random effect with their phylogenetic relatedness as part of the correlation structure. This can be done easily in metafor.

Response: As described above, we now incorporated phylogenetic relatedness into our mixed models by including a phylogenetic correlation structure into the random part of our mixed-effects models (see this revision).

Comment: L308-314: It is unclear to me from this description, the supplementary methods, and tables exactly what the model structures were and which models were compared with which using LRTs. The best I could gather (from Supplementary Table 2) is that a model with a single moderator/predictor (e.g. trophic group) was compared to a model that had two moderators and their interaction (e.g. trophic group x plant life form). But this is not a single-term deletion (the interaction and the single effect of plant life form have been removed together), so we cannot assess whether there is an interaction between life form and trophic group, or just a single effect of life form.

Response: We have revised the model structure. Namely, a model with a single moderator/predictor (e.g. trophic group) was compared to a model that had two moderators (e.g. trophic group + plant life form) and their interaction (e.g. trophic group x plant life form) (Supplementary Tables 2 and 3 in this revision). The effect of plant species with phylogenetic relatedness between plant species were treated as random effect in all the described models (lines 360-369).

Specifically, the R code was implemented in supplementary materials, and R script for these model comparisons were listed below:

Null model:

```
model1<-rma.mv(yi,vi,mods=~factor(Trophic)-1, random = list(~ 1 | Plant.species.new,~1|Plant.species.new.p), R=list(Plant.species.new.p=A), data=Total.data,method="ML")
```

Non-null models included two models, one was:

```
modelA<-rma.mv(yi,vi,mods=~factor(Trophic)+factor(life form)-1, random = list(~ 1 | Plant.species.new,~1|Plant.species.new.p),R=list(Plant.species.new.p=A),data=Total.data,method="ML")
```

The other was:

```
modelB<-rma.mv (yi, vi, mods = ~ factor (Trophic) + factor (life form)+factor(Trophic):factor(life form)-1, random = list(~ 1 | Plant.species.new,~1|Plant.species.new.p), R=list(Plant.species.new.p=A), data=Total.data,method="ML").
```

We used LRT to compare these models in this way:

```
anova(modelA,model1)
```

`anova(modelB,modelA)`

We have now re-written the corresponding section in the manuscript to make it more clear to readers.

Comment: L390-404: There is not enough information here and in Supplemental Table 3 to understand how the authors went about evaluating publication bias, even after looking at the referenced paper (Nakagawa and Santos; where several different approaches are discussed). Typical approaches include funnel plots (which would very helpful to add), and regression tests that test for funnel plot asymmetry. In these regression tests a significant p-value in these tests suggests publication bias. The p-values reported in Supp. Table 3 are highly significant, which may suggest strong publication bias (but I am not really sure what was done so this is not possible to evaluate).

Response: We thank the referee for bringing up this important point. We have now re-analysed our data to also account for publication bias, as suggested by the referee. As models of class "rma.mv" in R cannot be routinely screened for publication bias ("funnel.rma" and "regtest"), we used the sampling variances as moderator variables into our models to test for publication bias (lines 431-433). Further details can for example be found online on StackExchange

(<https://stats.stackexchange.com/questions/134768/metafor-package-in-r-ranktest-for-multivariate-meta-analysis>;

<https://stats.stackexchange.com/questions/155693/metafor-package-bias-and-sensitivity-diagnostics>)

Taking a null model as example, the relevant R code now becomes:

```
model1<-rma.mv(yi,vi,mods=~factor(Trophic)-1+vi, random = list(~ 1 |  
Plant.species.new,~ 1|Plant.species.new.p), R=list(Plant.species.new.p=A),  
data=Total.data,method="ML").
```

Regression test values and the corresponding p-value can be retrieved as:

```
c(tail(model1[["zval"]],1),tail(model1[["pval"]],1)) (see Supplementary Methods).
```

Comment: Fig. 2: Should the line between SMD plant genetic diversity and plant-damaging organism performance be red? Also, here and other figs: define SMD.

Response: The blue and red arrows denote positive and negative relationships, respectively, and we have changed the blue arrow into red arrow (Fig. 2). Also, we have defined SMD in all the figures. Namely, SMD is the abbreviation of Standardized Mean Difference (Figs. 2 and 3; Extended Data Figs. 22-24).

Comment: Fig. 3: It would be helpful if the widths of the lines were proportional to the strengths of the coefficients, so the reader can easily see which are the strongest relationships.

Response: We have added this (i.e., line width is proportional to the magnitude of the presented coefficient) in the figures (see the Figs. 2 and 3, and Extended Data Figs. 19-24

in this revision).

Comment: Extended data Fig. 1: It would be helpful to give the total number of comparisons/studies for each ecosystem type.

Response: We have presented the total number of studies for each ecosystem type in the figure caption of Extended Data Fig. 1— *a, In agroecosystems (335 studies). b, In grasslands (25 studies). c, In forests (15 studies). d, In old-field ecosystems (15 studies). e, In marine ecosystems (11 studies). f, In wetlands (10 studies). g, In shrublands (6 studies)* (see Extended Data Fig. 1 in this revision).

Comment: Extended data Fig. 5-18: I did not see that these figures were ever referenced or discussed in the MS at all. This seems problematic, especially because the results are counterintuitive based on the general hypotheses and broad patterns presented. In particular, as the intraspecific genetic diversity increases, we generally see that performance of plant-damaging organisms becomes more similar in the mixtures relative to monocultures (i.e. the effect size becomes less negative; Fig. 5a). This same general pattern holds for different types of plant antagonists and across different ecosystems, although there is some variability (Fig. 5-18).

Response: We have added a paragraph to present and discuss the results from Extended data Figs. 5-18 (lines 208-224).

Comment: Ext. Data Fig. 21: It is stated that arrow widths are proportional to coefficients but they are not—see comment above.

Response: Done as suggested (see above response).

Comment: Ext. Data Fig. 22: Natural enemies are mentioned in legend but not shown on figure.

Response: Natural enemies have been removed from the caption of Extended Data Fig. 22 (see Extended Data Fig. 22 in this revision). Results on natural enemies are contained in Fig. 3, showing the effects of SMD plant genetic diversity on tri-trophic interactions of invertebrate herbivore, natural enemy and plant performance.

Comment: Supplementary Table 1: Please add the plant species to this table, I think this is very important for the reader to get a sense of the breadth of plant species studied.

Response: We have added the plant species in Supplementary Table 1 (see Supplementary Table 1 in this revision).

Comment: Supplementary Table 2: Please rephrase description of response variable,

this is difficult to follow.

Response: Thanks for useful suggestion! We have rephrased description of response variable and re-analyzed the data again (see Supplementary Tables 2-16, Figs.1-3, Extended Data Figs. 1-24, and Supplementary Methods in this revision).

REVIEWER COMMENTS

Reviewer #1 (Remarks to the Author):

The manuscript is much improved and clarified. The revision exceeded my expectations.

Reviewer #2 (Remarks to the Author):

This is my second read of this paper. I am really excited about this topic and really impressed by the incredible dataset the authors have amassed. I also appreciate that the authors did a lot of changes in response to the past round of reviews that I think really strengthened the manuscript. I think the meta-analyses methods and models are more clear, and they added hypotheses and more biological interpretation that make the manuscript much more interesting. However, I still find it very challenging to interpret the methods and results, despite many hours spent looking at this paper and the long supplement. As I detail below, I could not reconcile the counterintuitive results in Fig. 5-18 with the SEMs. I think the SEMs appear simple to interpret on the surface, but the more one digs into the methods, the more challenging it becomes to understand. Ultimately, I felt I couldn't fully evaluate those results without more information, and although my default would be to assume the authors are correctly interpreting the patterns in their data, it seems plausible that the SEMs are misleading. Therefore, I think it would be essential to further explain those models and to show some of the actual values for SMD that are feeding into the SEMs.

L 117-118: Glad to see hypotheses added here, but Hypothesis 2 is a bit difficult to follow, consider bringing this into parallel structure with hypothesis 1, e.g. "plant genetic diversity increases plant performance indirectly by reducing the performance of plant antagonists and increasing the performance of natural enemies"

L 119-121: also consider rephrasing hypothesis 3: "the effects of plant genetic diversity on these multiple trophic levels and their trophic interactions will vary across different plant life forms, ecosystems and climatic zones"

L123-124: Does the study really address eco-evolutionary feedbacks??

L208-224: Its great to see that the authors added some description of the results in the extended data Fig. 5-18. However, I feel this description could be a bit misleading and there is no explanation for why the patterns here are counter to their hypotheses. They state that "the effect size of plant genetic diversity on plant antagonist performance significantly increased with the number of added genotypes" This makes it sound like the effect was getting stronger, but in fact the effect was disappearing because it was going from a clear negative effect to no effect. So, in other words, fields with two genotypes have significantly lower insect performance than monocultures, but in fields with 4 or 8 or 32 genotypes insect performance is the same as monocultures. I am quite perplexed by this outcome, and it is counter to their overarching hypothesis, but the authors don't offer any explanation for this. Is this a real biological pattern or a methodological artifact? Clearly sample sizes are decreasing as we increase in number of genotypes, so I wondered if that could be part of what is driving the pattern, but I somehow doubt that is all there is to it because it is quite consistent across different subsets of the data (in all cases the absolute magnitude of the effects diminish or don't change, they never increase with increasing diversity, which is what I would expect to see). So, I think there must be something real there, and if the authors also think this is a real biological pattern, it would be great to hear their thoughts on why this might be. In fact it could be one of the more interesting and surprising aspects of their study.

L347-350: I am confused by this statement that the "signs of SMDs in trophic groups" were converted to absolute values prior to analysis, but they considering that the reported SMD values vary from negative to positive.

L 367-369: Here the wording to me implies that a model with a single predictor was compared to a model that had two predictors plus an interaction term, which would make it challenging to understand the effects of any single term. However, if I am understanding Table S2 correctly it seems these were actually always single-term deletions. So, the model with trophic group + ecosystem type was compared to the base model with just trophic group, and the model with trophic group + ecosystem type + trophic group x ecosystem type would be compared to a model with trophic group + ecosystem type. If that is true I just suggest revising the text here to state that these were always single term deletions.

L 374: I thought I had a basic handle on these SEMs in the first read of the MS, but on this second review I spent even more hours staring at them and have left even more thoroughly confused. Does SMD genetic diversity represent the difference in number of genotypes between control and genetically diverse plots? So large values indicate the comparison was between monocultures and large numbers of genotypes (e.g. 32) and small numbers indicate the comparison was between monocultures and two genotypes?? And from what I can gather the values for "plant antagonist performance" and "plant performance" and "natural enemy performance" are also SMD values (based on L 386-387 and the R code on pg. 79 of the supplement). The challenge then is that interpreting the relationship of SMD plant genetic diversity and SMD herbivore performance between control and treatment is really challenging, especially because we never see the actual values of those numbers in the SEM plots (only the coefficients). For example, if we look at Extended Data Fig. 5a, this shows that the magnitude of the effects on herbivore performance are diminished as we increase in plant genetic diversity. But looking at Fig. 2, one would reasonably conclude the exact opposite pattern (that herbivores do worse as we increase diversity). I would like to assume that the authors are correctly interpreting their data that plant diversity decreases plant antagonist pressure, but I really cannot reconcile the patterns in Extended Data Figs 5-18 and the SEMs. It would be helpful to see simple regression plots of the individual relationships feeding into the SEMs, so we could see that actual data values (importantly, are they positive or negative).

Supplementary Table 1: Great the species were added here. Please fix the formatting so the column headers are at the start of each page

Replies to Reviewer #1:

Comment: The manuscript is much improved and clarified. The revision exceeded my expectations.

Response: We thank the reviewer for this positive evaluation.

Replies to Reviewer #2:

Comment: This is my second read of this paper. I am really excited about this topic and really impressed by the incredible dataset the authors have amassed. I also appreciate that the authors did a lot of changes in response to the past round of reviews that I think really strengthened the manuscript. I think the meta-analyses methods and models are more clear, and they added hypotheses and more biological interpretation that make the manuscript much more interesting.

Response: We thank the reviewer for this positive evaluation. We carefully addressed all comments in revising the manuscript. We are grateful for the suggestions, which have improved the quality of our study.

Comment: However, I still find it very challenging to interpret the methods and results, despite many hours spent looking at this paper and the long supplement.

Response: Thank you for requesting this additional information. We are sorry that these areas of the text were unclear. We have now included more detailed descriptions of Methods (lines 367-503) and Results (lines 125-296).

Comment: As I detail below, I could not reconcile the counterintuitive results in Fig. 5–18 with the SEMs. I think the SEMs appear simple to interpret on the surface, but the more one digs into the methods, the more challenging it becomes to understand. Ultimately, I felt I couldn't fully evaluate those results without more information, and although my default would be to assume the authors are correctly interpreting the patterns in their data, it seems plausible that the SEMs are misleading. Therefore, I think it would be essential to further explain those models and to show some of the actual values for SMD that are feeding into the SEMs.

Response: Thank you for identifying this specific location of the issue: the SEM. We have now explicitly added an own section in the Methods section in the main part of the manuscript to explain our two measures of plant genetic diversity. We are essentially dealing with two measures of plant genetic diversity: 1) a binary variable (0 = genotype monoculture, 1 = genotype mixture) that indicates either a genetically uniform monoculture or a genotype mixture (irrespective of how many genotypes were added); and 2) a

continuous variable that indicates the number of added plant genotypes over the control (expressed on a logarithmic basis); which further tests whether the number of genotypes has an effect. We have expanded on this in the Supplementary Methods section now, where we explain the two terms “plant genetic diversity” and “number of added plant genotypes” in detail (lines 367-372 and lines 416-428).

Comment: L 117-118: Glad to see hypotheses added here, but Hypothesis 2 is a bit difficult to follow, consider bringing this into parallel structure with hypothesis 1, e.g. "plant genetic diversity increases plant performance indirectly by reducing the performance of plant antagonists and increasing the performance of natural enemies".

Response: We thank the referee for these suggestions. We have revised the section according to the suggestions of the reviewer (lines 117-119).

Comment: L 119-121: also consider rephrasing hypothesis 3: "the effects of plant genetic diversity on these multiple trophic levels and their trophic interactions will vary across different plant life forms, ecosystems and climatic zones.

Response: We thank the referee for these suggestions. We followed the suggestion of the reviewer (lines 119-121).

Comment: L123-124: Does the study really address eco-evolutionary feedbacks??

Response: We have deleted the “*as well as eco-evolutionary feedbacks*” as this study did not really address eco-evolutionary feedbacks (lines 122-123).

Comment: L208-224: Its great to see that the authors added some description of the results in the extended data Fig. 5–18. However, I feel this description could be a bit misleading and there is no explanation for why the patterns here are counter to their hypotheses.

Response: In this revision, we have expanded to present the explanation for why the patterns here may seem to counter our hypotheses (lines 209-241). In the Extended Data Figs. 5–18, we tested the effects of “*number of added plant genotypes*” instead of the effects of “*plant genetic diversity*” on trophic groups. Namely, in the Extended Data Figs. 5–18, we explored the relationships between *the number of added plant genotypes* and the different effect sizes along with fitted meta-regression lines, while our hypotheses were mainly based on the responses of trophic groups to *plant genetic diversity* (Fig. 1b and Extended Data Figs. 1–4) and on the path analysis for the effects of SMD of plant genetic diversity on trophic interactions (Figs. 2 and 3; Extended Data Figs. 22–24).

Comment: They state that "the effect size of plant genetic diversity on plant antagonist performance significantly increased with the number of added genotypes". This makes it

sound like the effect was getting stronger, but in fact the effect was disappearing because it was going from a clear negative effect to no effect. So, in other words, fields with two genotypes have significantly lower insect performance than monocultures, but in fields with 4 or 8 or 32 genotypes insect performance is the same as monocultures. I am quite perplexed by this outcome, and it is counter to their overarching hypothesis, but the authors don't offer any explanation for this. Is this a real biological pattern or a methodological artifact? Clearly sample sizes are decreasing as we increase in number of genotypes, so I wondered if that could be part of what is driving the pattern, but I somehow doubt that is all there is to it because it is quite consistent across different subsets of the data (in all cases the absolute magnitude of the effects diminish or don't change, they never increase with increasing diversity, which is what I would expect to see). So, I think there must be something real there, and if the authors also think this is a real biological pattern, it would be great to hear their thoughts on why this might be. In fact it could be one of the more interesting and surprising aspects of their study.

Response: Thanks for your questions. We have now clarified this in lines 209-233.

Comment: L347-350: I am confused by this statement that the "signs of SMDs in trophic groups" were converted to absolute values prior to analysis, but they considering that the reported SMD values vary from negative to positive.

Response: Thank you for catching this error. We have replaced this sentence with "...the signs of SMDs in trophic groups were the original values for fitting the models" (line 390).

Comment: L 367-369: Here the wording to me implies that a model with a single predictor was compared to a model that had two predictors plus an interaction term, which would make it challenging to understand the effects of any single term. However, if I am understanding Table S2 correctly it seems these were actually always single-term deletions. So, the model with trophic group + ecosystem type was compared to the base model with just trophic group, and the model with trophic group + ecosystem type + trophic group x ecosystem type would be compared to a model with trophic group + ecosystem type. If that is true I just suggest revising the text here to state that these were always single term deletions.

Response: Your interpretation of these results is correct. We have clarified the text (lines 408-412).

Comment: L 374: I thought I had a basic handle on these SEMs in the first read of the MS, but on this second review I spent even more hours staring and them and have left even more thoroughly confused. Does SMD genetic diversity represent the difference in number of genotypes between control and genetically diverse plots? So large values indicate the

comparison was between monocultures and large numbers of genotypes (e.g. 32) and small numbers indicate the comparison was between monocultures and two genotypes?? And from what I can gather the values for "plant antagonist performance" and "plant performance" and "natural enemy performance" are also SMD values (based on L 386-387 and the R code on pg. 79 of the supplement). The challenge then is that interpreting the relationship of SMD plant genetic diversity and SMD herbivore performance between control and treatment is really challenging, especially because we never see the actual values of those numbers in the SEM plots (only the coefficients). For example, if we look at Extended Data Fig. 5a, this shows that the magnitude of the effects on herbivore performance are diminished as we increase in plant genetic diversity. But looking at Fig. 2, one would reasonably conclude the exact opposite pattern (that herbivores do worse as we increase diversity). I would like to assume that the authors are correctly interpreting their data that plant diversity decreases plant antagonist pressure, but I really cannot reconcile the patterns in Extended Data Figs 5–18 and the SEMs. It would be helpful to see simple regression plots of the individual relationships feeding into the SEMs, so we could see that actual data values (importantly, are they positive or negative).

Response: We apologize for our vague description. We have replaced “SMD plant genetic diversity” with “plant genetic diversity”. In this revision, we have explained the two terms “plant genetic diversity” and “number of added plant genotypes” in detail (lines 367-372 and lines 416-428). Fig. 1b, Fig. 2, Fig. 3, Extended Data Figs. 1–4 and Extended Data Figs. 22–24, were related with “plant genetic diversity”. Extended Data Figs. 5–18 and Extended data Figs. 19–21 were related with “the number of added plant genotypes” (see this revision).

Comment: Supplementary Table 1: Great the species were added here. Please fix the formatting so the column headers are at the start of each page.

Response: We have now added column headers at the start of each page (see Supplementary Table 1).

REVIEWERS' COMMENTS

Reviewer #3 (Remarks to the Author):

This manuscript reports on an interesting meta analysis that estimates effect sizes for how variation in plant genetic diversity alters interactions with plants, pathogens, herbivores, and upper trophic levels. There is great potential for this to be a high impact paper and it appears to be a substantial addition to existing meta analyses on this topic (Reiss/Drinkwater and Koricheva/Hayes, cited in the manuscript). I did not review an earlier version of the manuscript, but the responses to previous reviews were certainly sufficient. The paper could probably be published with just a few additional minor edits, but it is worth pointing out three main issues for consideration: some remaining statistical issues, a better explanation of the importance of estimated effect sizes, and some discussion about how phytochemistry is likely to mediate the effects of genetic diversity on ecological interactions. Revisions in response to these issues are not necessary, but the authors should consider them.

First, I hesitate to make comments on statistical analyses after the authors did a nice job expanding their explanations of the meta analysis and the piecewise structural equation models (really just directed acyclic graphs -DAGs - pieced together, since there were no latent variables included), so several of these are just points that could be ignored, but I hope that the authors find them to be useful considerations. It was fantastic to see path analysis combined with meta analysis, and it was also very nice to see that the package used for meta analyses (metafor) utilized a multilevel model, somewhat accounting for the pseudoreplication introduced by calculating multiple effect sizes for an individual paper. Here are some suggestions about this approach:

A. It is not clear how hierarchies are dealt with in the meta regression and piecewise path models, specifically how lower levels are matched when different hierarchies are available in different studies. The hierarchies are likely to be different (e.g., multiple species over multiple years in many plots within one study, and just a single version of all of these in another study). The consequences of this mismatches are hard to determine in a frequentist multilevel model, especially when linking DAGs into individual piecewise models and then comparing those models and making inferences about the differences among those models. A clear solution to this would be a hierarchical Bayesian model, which would take estimates from lower levels in the hierarchy for a given study as priors and yield posterior estimates for each study that would then enter the multiple regression or piecewise path model in a more standardized way. The piecewise path models are nice because they are not subject to the large sample size requirements of full SEMs, so using fewer data points (i.e. one value per study that comes out of the hierarchical Bayesian model) would yield more accurate path coefficients and would allow for fairer comparisons among models.

B. The piecewise path models do have some advantages, but they are not SEM models in the sense that there are no latent variables, which is fine, but these data really do have a clear latent variable component, as alluded to multiple times in the manuscript. For example, "plant performance" is an unmeasurable latent variable that is estimated by measured variables, including various measures of growth, reproduction, and quality, so an SEM would include performance as a latent variable that causes (arrows point to) those multiple measured variables (and this would allow for including all of the variable measures of growth, reproduction, and quality as measured variables in a traditional SEM). The resulting estimated path coefficients are likely to be more similar among the different models tested using this approach as well as the suggestion in A above (hierarchical Bayesian model).

C. The focus on statistical significance (mentioned 30 times in the manuscript) is distracting, why not just report the effect sizes, p-values, and the authors' own inferences? (See the 2016 ASA statement on p-values and many subsequent studies in ecology, psychology, and clinical studies).

D. Why just use Hedges G and not other measures, such as log ratio or measures of variance (see Nakagawa et al., *Methods in Ecology and Evolution* 2015, 6, 143–152 for an example of the latter)? Some justification would be helpful.

Second, it would be helpful to this reader if the authors added a little about how to interpret effect sizes (e.g., report examples of raw effect sizes from the actual papers and mention how they are meaningful) and to put them into perspective as much as possible. My use of "effect size" here refers to magnitude of parameter estimates from the individual studies rather than just the Hedges G (the SMD equation in this paper) summary effect size of all the studies, such as differences in the response variables between levels of independent variables (or between "control" and "experimental" levels), raw regression slopes, raw differences in variances, and other such easily interpretable values. For example, just a few strategic examples highlighting differences in numbers of herbivores on clones versus mixed varieties and similar such values would help put the standardized effect sizes into ecological perspective. It is also difficult to make the connection from changes in response variables like "herbivore performance" and ideas about how herbivore population dynamics might change in response to genetic diversity, so it would help to include more on how those connections are made as well as acknowledgement of inferential pitfalls in over-extrapolating relatively small effect sizes.

Third, chemical profiles of plants are extremely important functional traits that have measurable ecological and evolutionary effects on species interactions for most primary producers and in most ecosystems, including algae, macrophytes, terrestrial plants, and endophytes. It is also known that these chemical profiles vary quite dramatically within a species (this is particularly well studied in agricultural systems). Thus, it was surprising that the roles of chemistry in mediating effects of genetic diversity on interactions was mentioned only once (line 186 on allelopathy). There are documented effects of intraspecific variation in plant chemistry on a variety of interactions, but this is an area of study that should be expanded, and a discussion of these mechanistic pathways could increase the impact of this synthesis paper substantially (for examples of relevant references, see Hunter, 2016, *The Phytochemical Landscape*).

MINOR specific comments:

Lines 35-36: change "in different plant life forms" to "for different plant life forms"

Lines 50-51: "can also be obtained by.." – it's not clear what this means. Genetic diversity can be manipulated by modifying richness? Variation in genetic diversity can be obtained by manipulations?

Lines 52-55: because this is all mediated by chemistry, it is a huge shortcoming to leave out how much chemistry can vary within a species and how that intraspecific variation affects interacting species. This also comes up again in lines 68-73, where a lot of this is mediated through plant chemistry - both primary and secondary metabolites are key mechanisms driving direct and indirect effects on upper trophic levels. Another example of this issue is in lines 152-181, where it is a great omission to leave out plant chemistry.

Lines 81-84: I think this meta analysis provides significant advances from Reiss/Drinkwater and Koricheva/Hayes meta analyses, but a little more about those previous findings would help here and in the discussion.

Line 235: change to "the number of added plant genotypes"

Lines 240-242: here and elsewhere (e.g., tropical systems), "small sample sizes" are alluded to being responsible for "no significant differences," but why report these results at all if the sample sizes are insufficient for good estimates of effect sizes? Or why not get rid of the term "significance" and just show confidence or credible intervals? See comments above.

Lines 321-324 and lines 330-333: these issues could be efficiently managed with a hierarchical Bayesian model (see comments above) rather than the variable mix of approaches employed here (different types of nesting in multilevel models). The fact is that no effect sizes within a study are independent (there is plenty of justification for this comment, but also see Nakagawa and Santos for an example, *Evolutionary Ecology*, 26, 1253-1274), so this issue needs to be dealt with in a more standardized way for all analyses.

Lines 375-379: Based on the equation for SMD, this is just Hedges G - why not call it that? Also, add a line of justification for why this is used instead of Cohen's D, log ratio, or variance estimates (see comments above).

Replies to Reviewer #3:

Comment: This manuscript reports on an interesting meta analysis that estimates effect sizes for how variation in plant genetic diversity alters interactions with plants, pathogens, herbivores, and upper trophic levels. There is great potential for this to be a high impact paper and it appears to be a substantial addition to existing meta analyses on this topic (Reiss/Drinkwater and Koricheva/Hayes, cited in the manuscript). I did not review an earlier version of the manuscript, but the responses to previous reviews were certainly sufficient. The paper could probably be published with just a few additional minor edits, but it is worth pointing out three main issues for consideration: some remaining statistical issues, a better explanation of the importance of estimated effect sizes, and some discussion about how phytochemistry is likely to mediate the effects of genetic diversity on ecological interactions. Revisions in response to these issues are not necessary, but the authors should consider them.

Response: We thank the reviewer for this positive evaluation. We carefully addressed all comments in revising the manuscript. We are grateful for the suggestions, which have improved the quality of our study (see this revision).

Comment: First, I hesitate to make comments on statistical analyses after the authors did a nice job expanding their explanations of the meta analysis and the piecewise structural equation models (really just directed acyclic graphs -DAGs - pieced together, since there were no latent variables included), so several of these are just points that could be ignored, but I hope that the authors find them to be useful considerations. It was fantastic to see path analysis combined with meta analysis, and it was also very nice to see that the package used for meta analyses (metafor) utilized a multilevel model, somewhat accounting for the pseudoreplication introduced by calculating multiple effect sizes for an individual paper. Here are some suggestions about this approach:

Response: We thank the reviewer for this positive evaluation. We carefully addressed all comments in revising the manuscript. We are grateful for the suggestions, which have improved the quality of our study (see this revision).

Comment: A. It is not clear how hierarchies are dealt with in the meta regression and piecewise path models, specifically how lower levels are matched when different hierarchies are available in different studies. The hierarchies are likely to be different (e.g., multiple species over multiple years in many plots within one study, and just a single version of all of these in another study). The consequences of this mismatches are hard to determine in a frequentist multilevel model, especially when linking DAGs into individual piecewise models and then comparing those models and making inferences about the

differences among those models. A clear solution to this would be a hierarchical Bayesian model, which would take estimates from lower levels in the hierarchy for a given study as priors and yield posterior estimates for each study that would then enter the multiple regression or piecewise path model in a more standardized way. The piecewise path models are nice because they are not subject to the large sample size requirements of full SEMs, so using fewer data points (i.e. one value per study that comes out of the hierarchical Bayesian model) would yield more accurate path coefficients and would allow for fairer comparisons among models.

Response: Thanks for your comments. We absolutely agree with you that it can be challenging to find an optimal solution for dealing with hierarchically structured data in the context of meta-regressions and piecewise path analyses. For considering hierarchies in a frequentist approach to multilevel modeling, we considered 1) each study as a random intercept term in the mixed-effect model; 2) the different plant species as random intercepts, nested within study; and 3) the phylogenetic relatedness between plant species as part of the correlation structure (Lines 434-439). We are aware that a Bayesian approach would certainly also have been a useful avenue for us to take, but ideally both frequentist and Bayesian approaches should ultimately lead to similar conclusions.

Comment: B. The piecewise path models do have some advantages, but they are not SEM models in the sense that there are no latent variables, which is fine, but these data really do have a clear latent variable component, as alluded to multiple times in the manuscript. For example, “plant performance” is an unmeasurable latent variable that is estimated by measured variables, including various measures of growth, reproduction, and quality, so an SEM would include performance as a latent variable that causes (arrows point to) those multiple measured variables (and this would allow for including all of the variable measures of growth, reproduction, and quality as measured variables in a traditional SEM). The resulting estimated path coefficients are likely to be more similar among the different models tested using this approach as well as the suggestion in A above (hierarchical Bayesian model).

Response: We thank the referee for these stimulating comments. It is true that “plant performance” is a mental concept that could have been incorporated in a structural equation model as a latent variable. The same is true, however, for many other variables we used in our study (see detailed description in the Supplementary Information – “Further description of predictor variables”), and we deliberately decided to use meta-regression approaches rather than a latent variable approach (as latent variable models often show convergence or specification problems). In our case, “plant performance” included growth, reproduction and quality of plants (Lines 104-105), and different categories (i.e., growth, reproduction and quality of plants) covered distinct plant species. We acknowledge that a

latent variable approach would have been interesting, too – but this would have considerably altered the overall approach used in this manuscript.

Comment: C. The focus on statistical significance (mentioned 30 times in the manuscript) is distracting, why not just report the effect sizes, p-values, and the authors' own inferences? (See the 2016 ASA statement on p-values and many subsequent studies in ecology, psychology, and clinical studies).

Response: We agree that P values should be abandoned and we are aware of the statement by the ASA. We admit that only focusing on statistical significance as seen from p-values is not the best way to explain the results. To address the referee's comment, we have now added the effect sizes, along with p-values and other statistics in this revision. We hope that this addresses the comment sufficiently. In addition, we have added Supplementary Table 16, where we list the statistical tests for the relationship between number of added genotypes in the plant genetic diversity treatment over the control and the different effect sizes along with fitted meta-regression lines (See Supplementary information).

Comment: D. Why just use Hedges G and not other measures, such as log ratio or measures of variance (see Nakagawa et al., *Methods in Ecology and Evolution* 2015, 6, 143–152 for an example of the latter)? Some justification would be helpful.

Response: Thank you for your comments. Several measures for calculating the effect size have been established to evaluate the difference between two different groups of a certain variable. Here, we adopted Hedges G (Lines 414-420) because Hedges G is a very popular measure for calculating effect size. For example, *Nature*, 555(7695):175-182 (2018). doi: 10.1038/nature25753; *JAMA*, 298(4):430-7 (2007). doi: 10.1001/jama.298.4.430; *BMJ*, 339:b3128 (2009). doi: 10.1136/bmj.b3128.

Comment: Second, it would be helpful to this reader if the authors added a little about how to interpret effect sizes (e.g., report examples of raw effect sizes from the actual papers and mention how they are meaningful) and to put them into perspective as much as possible. My use of "effect size" here refers to magnitude of parameter estimates from the individual studies rather than just the Hedges G (the SMD equation in this paper) summary effect size of all the studies, such as differences in the response variables between levels of independent variables (or between "control" and "experimental" levels), raw regression slopes, raw differences in variances, and other such easily interpretable values. For example, just a few strategic examples highlighting differences in numbers of herbivores on clones versus mixed varieties and similar such values would help put the standardized effect sizes into ecological perspective. It is also difficult to make the connection from

changes in response variables like “herbivore performance” and ideas about how herbivore population dynamics might change in response to genetic diversity, so it would help to include more on how those connections are made as well as acknowledgement of inferential pitfalls in over-extrapolating relatively small effect sizes.

Response: Thanks for your comments. We have now altered the text accordingly and added a section on how we defined effect sizes in the present study (Lines 407-413). Definition of effect size and its measures: In the context of the present study, effect size is defined for each study or observation, as the sign and magnitude of parameter estimates from the individual studies or observations. In addition, when relationships between one variable and another were considered, effect sizes were also represented by the differences in the response variables between levels of independent variables (or between “control” and “experimental” levels) or by raw regression slopes between the response variables and the continuous predictors.

Comment: Third, chemical profiles of plants are extremely important functional traits that have measurable ecological and evolutionary effects on species interactions for most primary producers and in most ecosystems, including algae, macrophytes, terrestrial plants, and endophytes. It is also known that these chemical profiles vary quite dramatically within a species (this is particularly well studied in agricultural systems). Thus, it was surprising that the roles of chemistry in mediating effects of genetic diversity on interactions was mentioned only once (line 186 on allelopathy). There are documented effects of intraspecific variation in plant chemistry on a variety of interactions, but this is an area of study that should be expanded, and a discussion of these mechanistic pathways could increase the impact of this synthesis paper substantially (for examples of relevant references, see Hunter, 2016, *The Phytochemical Landscape*).

Response: We thank the referee for this idea. We have now cited the following five references to discuss the effects of chemical profiles of plants on trophic groups and trophic interactions.

1. Hunter, M. D. *The Phytochemical Landscape: Linking Trophic Interactions and Nutrient Dynamics*. Princeton University Press (2006).
2. Degen, T., Dillmann, C., Marion-Poll, F. & Turlings, T. C. J. High genetic variability of herbivore-induced volatile emission within a broad range of maize inbred lines. *Plant Physiol.* **135**, 1928–1938 (2004).
3. Zytynska, S. E. et al. Effect of plant chemical variation and mutualistic ants on the local population genetic structure of an aphid herbivore. *J. Anim. Ecol.* **88**, 1089–1099 (2019).
4. Züst, T. & Agrawal, A. A. Plant chemical defense indirectly mediates aphid performance via interactions with tending ants. *Ecology* **98**, 601–607 (2017).
5. Senft, M., Clancy, M. V., Weisser, W. W., Schnitzler, J. P. & Zytynska, S. E. Additive effects of plant chemotype, mutualistic ants and predators on aphid performance and survival. *Funct. Ecol.* **33**, 139–151

(2019).

The added sentences were presented in this revision (Lines 55-57, 71-75, 183-186, 191-192, 202-203, 206-207, and 217-219).

Comment: MINOR specific comments: Lines 35-36: change “in different plant life forms” to “for different plant life forms”.

Response: We have changed “in different plant life forms” to “for different plant life forms” (Line 36).

Comment: Lines 50-51: "can also be obtained by.." – it's not clear what this means. Genetic diversity can be manipulated by modifying richness? Variation in genetic diversity can be obtained by manipulations?

Response: Done as suggested (Lines 50-51).

Comment: Lines 52-55: because this is all mediated by chemistry, it is a huge shortcoming to leave out how much chemistry can vary within a species and how that intraspecific variation affects interacting species. This also comes up again in lines 68-73, where a lot of this is mediated through plant chemistry - both primary and secondary metabolites are key mechanisms driving direct and indirect effects on upper trophic levels. Another example of this issue is in lines 152-181, where it is a great omission to leave out plant chemistry.

Response: We thank the referee for this suggestion. We have added several sentences and references to introduce and discuss the effects of intraspecific variation of plant chemistry on trophic groups and trophic interactions (Lines 55-57, 71-75, 183-186, 191-192, 202-203, 206-207, and 217-219).

Comment: Lines 81-84: I think this meta analysis provides significant advances from Reiss/Drinkwater and Koricheva/Hayes meta analyses, but a little more about those previous findings would help here and in the discussion.

Response: We thank the referee for this suggestion. We have added several sentences to introduce and discuss those previous findings of Reiss/Drinkwater and Koricheva/Hayes meta analyses (Lines 80-84 in the Introduction section; Lines 314-321 in the Discussion section).

Comment: Line 235: change to "the number of added plant genotypes"

Response: Done as suggested (Line 254).

Comment: Lines 240-242: here and elsewhere (e.g., tropical systems), "small sample

sizes" are alluded to being responsible for "no significant differences," but why report these results at all if the sample sizes are insufficient for good estimates of effect sizes? Or why not get rid of the term "significance" and just show confidence or credible intervals? See comments above.

Response: Thank you! We admit that only focusing on statistical significance due to p-value is not the best way to explain the results. Therefore, to enhance the readability of our manuscript, we have added the effect sizes, p-values, and other statistic value in this revision. In addition, we have added a Supplementary Table 16 to list the statistic values for the relationship between number of added genotypes in the plant genetic diversity treatment over the control and the different effect sizes along with fitted meta-regression lines (Lines 127-313; Supplementary information).

Comment: Lines 321-324 and lines 330-333: these issues could be efficiently managed with a hierarchical Bayesian model (see comments above) rather than the variable mix of approaches employed here (different types of nesting in multilevel models). The fact is that no effect sizes within a study are independent (there is plenty of justification for this comment, but also see Nakagawa and Santos for an example, *Evolutionary Ecology*, 26, 1253–1274), so this issue needs to be dealt with in a more standardized way for all analyses.

Response: Thanks for your comments. We absolutely agree with your suggested solution for dealing with hierarchies you mentioned. For considering hierarchies in a frequentist multilevel model, to diminish hierarchies, we considered 1) each study as random effect in the mixed-effect model; 2) the different plant species as random effects, nested within study, in the mixed-effect model; and 3) the correlations between any two plant species as an additional part of the random part in the mixed-effect model (Lines 434-439).

Comment: Lines 375-379: Based on the equation for SMD, this is just Hedges G - why not call it that? Also, add a line of justification for why this is used instead of Cohen's D, log ratio, or variance estimates (see comments above).

Response: Thank you for your comments. Several measures for calculating the effect size have been established to evaluate the difference between two different groups of a certain variable. Here, we adopted Hedges G (Lines 414-420) because Hedges G is a very popular measure for calculating effect size. For example, *Nature*, 555(7695):175-182 (2018). doi: 10.1038/nature25753; *JAMA*, 298(4):430-7 (2007). doi: 10.1001/jama.298.4.430; *BMJ*, 339:b3128 (2009). doi: 10.1136/bmj.b3128.